# Unleashing Graph Transformers with Green and Martin Kernels

## Abstract

Graph Transformers (GTs) are rapidly emerging as superior models, surpassing traditional message-passing neural networks in graph-level tasks. For optimal performance, it is essential to design GT architectures that embed graph inductive biases and utilize global attention mechanisms through effective structural encodings (SEs). In this work, we introduce novel SEs derived from a rigorous theoretical analysis of random walks (RWs), specifically leveraging the Green and Martin kernels. The Green and Martin kernels are mathematical tools used to observe the long-term behavior of RWs on graphs. By integrating these kernels into the encoding process, we enhance their capability to accurately represent complex graph structures. Our empirical evaluations demonstrate that these approaches enable GTs to achieve state-of-the-art performance on 7 out of 8 benchmark datasets. These include molecular datasets characterized by intricate, non-aperiodic substructures such as benzene rings, and directed acyclic graphs common in the circuit domain. We attribute these performance improvement to the effective capture of the characteristics of non-aperiodic substructures and directed acyclic graphs by our extending encodings. The results not only validate the effectiveness of integrating the Green and Martin kernels into RW-based encodings but also underscore their potential to substantially enhance the learning capabilities of GTs across diverse applications.

## 1 Introduction

Graph Transformers (GTs) (1; 2; 3; 4; 5; 6) have been proposed as a superior alternative to conventional Message Passing Neural Networks (MPNNs) (7), mitigating MPNNs' well-known issues such as over-smoothing (8; 9), over-squashing (10; 11), under-reaching (12; 13), and limited expressive power (14; 15). For GTs to perform effectively, it is essential to incorporate inductive biases (16) specific to graph data. Additionally, global attention of GTs requires structural encodings (SEs) that enable precise differentiation of nodes within the graph and its substructures (17). Recent model, GRIT (6), has shown state-of-the-art (SOTA) performance for various benchmarks by implicitly incorporating the graph inductive bias from message-passing and the global attention advantages of Transformer (18) by using Relative Random Walk Probabilities (RRWP) as the SE.

Random walks (RWs) have been extensively studied as a means of exploring the structure of graphs. Several results (19; 20; 21) have demonstrated that the long-term behavior of a RW encapsulates the topological information of the graph. There are two mathematical quantities that capture this long-term behavior, providing valuable insights into the graph's characteristics. The *Green kernel* is classically defined as a "pseudo-inverse operator" used to find solutions to equations (22). Analogously, the Green kernel on a graph can be defined by the Moore-Penrose Inverse of the graph Laplacian matrix in (23; 24), in which the Green kernel is utilized for calculating various probabilistic quantities. In the context of RWs on graphs, it represents the expected number of visits to one node from another (25). The *Martin kernel*, on the other hand, is an important function in probability theory and potential theory, particularly within the framework of Martin boundary theory (26). By utilizing Martin kernel, one can construct the harmonic potential function, which is essential tool for analyzing behavior of the Brownian motion (27). The Martin kernel is defined on graphs analogously and used for constructing the harmonic potential function (20). Moreover, it is equal to the probability that a RW starting from one node will reach another node within a finite time (21).

In this study, we introduce two extended concepts from the theory of RWs—the Green and Martin kernels (21; 25; 28)—as new SEs. To the best of our knowledge, our paper is the first to introduce these concepts to design SEs. We apply these new SEs to the recently developed GRIT model (6) and demonstrate that they outperform existing methods on various benchmark datasets, including the molecular and circuit domains. Through this targeted innovation, we present new SEs that are not only specialized for non-aperiodic substructures and DAGs, but also enhanced performance across diverse benchmarks.

In summary, our contributions are as follows. (i) We propose new SEs, *Green Kernel Structural Encoding* (GKSE) and *Martin Kernel Structural Encoding* (MKSE), based on the Green and Martin kernels, which extend the RWs. (ii) We integrate GKSE and MKSE into the GRIT, demonstrating that GKSE and MKSE surpass SOTA performance across various graph benchmarks, including those with small, medium, large, and long-range interactions. (iii) In molecular graphs containing numerous non-aperiodic substructures (e.g., benzene rings), our methods demonstrate efficient learning and strong performance on ZINC (29) and PCQM4Mv2 (30). (iv) We demonstrate that GKSE and MKSE efficiently learn circuit domain data, which is represented as a DAG, and provide several baseline benchmark results on Open Circuit Benchmark (OCB), the first graph benchmark dataset in the circuit domain.

## 2 RELATED WORK

**Graph Transformers.** Graph Neural Networks (GNNs) have advanced considerably, evolving from MPNNs (7) to sophisticated GTs. GTs capitalize on Transformers' flexibility and scalability, incorporating SEs to enhance learning from graph data. Notably, NAGphormer (31), AGT (32) and TokenGT (33) leverage Laplacian eigenvectors as SEs, demonstrating strong performance in node classification tasks by effectively capturing the global structure of graphs. In addition, the magnetic Laplacian have been explored for SEs, emphasizing the importance of directed graphs (34).

GT models have also evolved to incorporate edge attributes, improving their abbility to capture the structural information present in graphs. The Graphormer (1) and EGT (2) enhance the self-attention mechanism by integrating edge attributes, which improve the interaction between node attributes during the learning process. Moreover, several models have introduced relative SEs to handle edge attributes more effectively. GraphGPS (5) applies relative SEs to facilitate interaction between local message-passing and global self-attention mechanism. In particular, the GRIT (6) employs a multimodal approach to incorporating both node and edge attributes into the self-attention mechanism. It achieves high performance by utilizing RRWP, an SE based on RWs, effectively capturing the structural properties of graphs. For comprehensive insights into GTs, readers can refer to detailed surveys that cover recent methodologies, challenges, and future research opportunities (35; 36).

**Structural Encodings for Graphs.** GTs encounter notable challenges in encoding structural information, which are crucial for distinguishing non-isomorphic structures and utilizing graph symmetries. In this paper, we consider and utilize SE as node and edge representations that are invariant to graph isomorphisms, in order to better capture the structural information of the graph (37).

For SEs, the use of graph Laplacians in graph analysis has been widely explored. One study introduced globally consistent anisotropic kernels using Laplacian eigenvectors to incorporate directional information in GNNs (38). Additionally, other research has generalized graph Laplacians, demonstrating their effectiveness in capturing the geometric structure of graphs (17; 39). Researchers have tackled the constraints of spectral methods through the development of SignNet and BasisNet, which maintain invariance to sign flips and the basis symmetries of eigenvectors (40). Additionally, another study leveraged the eigenvectors of the Magnetic Laplacian to integrate directional information into SEs (34; 41). Furthermore, an alternative approach utilized the Hodge 1-Laplacian spectrum for creating edge-level SEs (42). On the other hand, the Random Walk-based Structural Encoding (RWSE) has been proposed (43), while direction- and structure-aware SEs for directed graphs based on directional RWs have been developed (34). In addition, the RRWPs were proposed using RW probabilities and learned relative SEs (6). This was extended by applying edge-level RWs on a simplicial complex for edge SEs in graphs (42).

## 3 Mathematical Background

The Green and Martin kernels are mathematical tools that capture the long-term behavior of RWs on graphs. Specifically, both kernels are functions of node pairs, and from this perspective, we utilize them as absolute or relative SEs for GNNs or GTs. When used as SEs, these kernels leverage the RW information that inherently reflects the topological properties of the graph, enabling the model to better capture structural patterns. We begin by describing RWs on grpahs as a stochastic process, focusing on the Green and Martin kernels.

### 3.1 Random Walk on Graphs as a Stochastic Process

Let $\mathcal{G} = (\mathcal{V}, \mathcal{E})$ be a graph, and let $\mathbf{P} : \mathcal{V} \times \mathcal{V} \to \mathbb{R}$ represent the transition probability kernel of $\mathcal{G}$, that is, $\mathbf{P}(x, y)$ denotes the probability that a RW starting at node $x$ moves to node $y$ in the next step. We define $\mathbf{P}^{(i)} : \mathcal{V} \times \mathcal{V} \to \mathbb{R}$ for $i \in \mathbb{N}$, by performing the convolution of $\mathbf{P}$ with itself as follows: for $x, y \in \mathcal{V}$,

$$\mathbf{P}^{(i)}(x, y) = \int_{\mathcal{V}} \mathbf{P}^{(i-1)}(x, z) \, \mathbf{P}(z, y) \, dz. \tag{1}$$

We note that if $\mathcal{G}$ is a finite graph with $n$ nodes, $\mathbf{P}$ can be represented as an $n \times n$ matrix defined by $\mathbf{P} = \mathbf{D}^{-1}\mathbf{A}$, where $\mathbf{D}$ is the diagonal matrix with the degrees of the nodes as its diagonal entries, and $\mathbf{A}$ is the adjacency matrix of the graph $\mathcal{G}$. In this case, $\mathbf{P}^{(i)}$ is the matrix $\mathbf{P}^i$, which is obtained by multiplying $\mathbf{P}$ by itself $i$ times.

We define the sequence spaces $\Omega(x) \subset \mathcal{V}^{\mathbb{N} \cup \{0\}}$ for $x \in \mathcal{V}$ as follows:

$$\Omega(x) = \left\{ \omega = (x, \omega_1, \omega_2, \dots) \in \mathcal{V}^{\mathbb{N} \cup \{0\}} \mid (x, \omega_1), (\omega_i, \omega_{i+1}) \in \mathcal{E}, \ \forall i \in \mathbb{N} \right\}, \tag{2}$$

that is, the space of all forward trajectories derived from the RW on $\mathcal{G}$ starting from the node $x$. We simply denote by $\Omega = \cup_{x \in \mathcal{V}} \Omega(x)$, which is the space of all forward trajectories.

We define the probability measure $\mathbb{P}_{\omega \in \Omega(x)}$ on $\Omega(x)$ such that, for $u_1, \dots, u_k \in \mathcal{V}$,

$$\mathbb{P}_{\omega \in \Omega(x)}(\{\omega \in \Omega(x) \mid \omega_i = u_i, \ \forall i = 1, \dots, k\}) := \mathbf{P}(x, u_1) \cdot \prod_{i=1}^{k-1} \mathbf{P}(u_i, u_{i+1}), \tag{3}$$

which means the probability that a RW starting at node $x$ passes through $u_1, \dots, u_k$ in that specific order.

Lastly, we define the set of random variables $X = \{X_i : \Omega \to \mathcal{V}\}_{i \in \mathbb{N} \cup \{0\}}$ by $X_i(\omega) = \omega_i$ for all $i \in \mathbb{N} \cup \{0\}$ and $\omega \in \Omega$. We intentionally omit further mathematical details, such as $\sigma$-algebra and precise construction of measure, for the sake of simplicity. For a more detailed explanation, please refer to (44).

The triple $(\Omega, (\mathbb{P}_{\omega \in \Omega(x)})_{x \in \mathcal{V}}, X)$ uniquely determines the RW on a graph. For example, for $x, y \in \mathcal{V}$ and $i \in \mathbb{N} \cup \{0\}$, the probability that a RW starting from $x$ will visit $y$ after $i$ step is $\mathbb{P}_{\omega \in \Omega(x)}[X_i(\omega) = y]$. We note that the transition probability matrix $\mathbf{P}$ also uniquely determines the RW. Observe that the kernel $\mathbf{P}^{(i)}$ represents the probability moving from one node to another node in $i$ steps. In other word, the value $\mathbf{P}^{(i)}(x, y)$ is equal to $\mathbb{P}_{\omega \in \Omega(x)}[X_i(\omega) = y]$.

### 3.2 Green Kernel and Martin Kernel on Graphs

In this subsection, we introduce the *Green kernel* and *Martin kernel*, which are essential tools in understanding RWs on graphs. The Green kernel represents the expected number of visits from one node to another, while the Martin kernel describes the probability of reaching a specific node from another within a finite number of steps. One can formulate both kernels as follows: for $x, y \in \mathcal{V}$,

$$\text{(Green kernel)} \quad G(x, y) = \mathbb{E}_{\omega \in \Omega(x)}[L_y^{(\infty)}(\omega)]; \tag{4}$$

$$\text{(Martin kernel)} \quad M(x, y) = \mathbb{P}_{\omega \in \Omega(x)}[\tau_y(\omega) < \infty], \tag{5}$$

where $L_y^{(k)} : \Omega \to \mathbb{N} \cup \{0\}$ is the *counting function* and $\tau_y : \Omega \to \mathbb{N} \cup \{0\}$ is the *first hitting time map*, which are defined as follows: for $\omega \in \Omega$,

$$L_y^{(k)}(\omega) = \sum_{i=0}^{k} \mathbb{1}_{\{X_i(\omega)=y\}}; \tag{6}$$

$$\tau_y(\omega) = \min\{i \in \mathbb{N} \cup \{0\} \mid X_i(\omega) = x\}. \tag{7}$$

Both kernels are deeply connected to the underlying structure of the graph, as they reflect important topological properties of the graph (21; 25; 20). However, it is important to note that the Green and Martin kernels are primarily meaningful in transient graphs, where RWs do not return to the starting node infinitely often. In fact, for a recurrent graph, which is a non-transient graph, the value of Green kernel is always $+\infty$ and the value of Martin kernel remains constantly 1.

### 3.3 Adapting Green and Martin Kernels for Recurrent Graphs

Most graph data in practical applications tends to be recurrent rather than transient, which makes the computation of the traditional Green and Martin kernels less meaningful. In finite graphs, RWs are recurrent unless there is a sink region that terminates the walk (i.e., a killed process). This issue arises because the traditional Green and Martin kernels capture the long-term behavior of RWs over infinite time, where RWs repeatedly revisit nodes. To address this limitation, we developed new versions of the Green and Martin kernels by restricting the RW to a finite number of steps. This approach allows us to create new kernels for our proposed SEs, which reflect meaningful RW properties even in recurrent graphs by capturing the behavior over a finite horizon.

One further issue is that, for all $x \in \mathcal{V}$, the value of the Martin kernel at $(x, x)$ is always 1 because the RW immediately revisits itself at step 0. As a result, even when using the finite-step Martin kernel, the absolute SE consists of constant 1s. To resolve this, we replace the first hitting time map $\tau_x$ with the first return time map $\tau_x^+ : \Omega \to \mathbb{N} \cup \{0\}$, which is defined as

$$\tau_x^+(\omega) := \min\{i \in \mathbb{N} \mid X_i(\omega) = x\}, \quad \forall \omega \in \Omega. \tag{8}$$

By modifying the definition of the Martin kernel to use the first return time map, the absolute SE reflects the topology of the graph. More specifically, it becomes the probability that a RW starting from a node returns to itself. This adjustment ensures that the SE reflects more meaningful information about the graph structure. Importantly, the relative SE remains unaffected by this change.

## 4 Methodology: Introducing GKSE and MKSE

In this section, we introduce our proposed *Green Kernel Structural Encoding* (GKSE) and *Martin Kernel Structural Encoding* (MKSE), which are designed to reflect the theoretical significance of the Green and Martin kernels discussed in the previous section. Moreover, these encodings incorporate all the considerations discussed in section 3.3, leading to the development of new mathematical constructs that effectively capture meaningful structural properties of graphs.

### 4.1 Green and Martin Kernel Structural Encodings

Applying observations in previous section 3.3, we now introduce our GKSE and MKSE. First, we define the finite-step Green kernel and finite-step Martin kernel to capture meaningful RW behavior within a limited number of steps, whose meanings are as follows: for $x, y \in \mathcal{V}$ and $k \in \mathbb{N} \cup \{0\}$,

$$\text{(finite-step Green kernel)} \quad \mathbb{E}_{\omega \in \Omega(x)}[L_y^{(k)}(\omega)]; \tag{9}$$

$$\text{(finite-step Martin kernel)} \quad \mathbb{P}_{\omega \in \Omega(x)}[\tau_y^+(\omega) \le k]. \tag{10}$$

Mathematically, the finite-step Green kernel represents the expected number of visits from one node to another within $k$ steps, while the finite-step Martin kernel approximates the probability of reaching a specific node from another within $k$ steps.

Based on these definitions, we construct the GKSE : $\mathcal{V} \times \mathcal{V} \to \mathbb{R}^K$ and MKSE : $\mathcal{V} \times \mathcal{V} \to \mathbb{R}^K$ with the dimension $K$ of SE as follows: for $x, y \in \mathcal{V}$,

$$\text{GKSE}\,(x,y) = [\mathbf{G}^{(0)}(x,y), \mathbf{G}^{(1)}(x,y), \dots, \mathbf{G}^{(K-1)}(x,y)]; \tag{11}$$

$$\text{MKSE}\,(x,y) = [\mathbf{M}^{(0)}(x,y), \mathbf{M}^{(1)}(x,y), \dots, \mathbf{M}^{(K-1)}(x,y)], \tag{12}$$

where $\mathbf{G}^{(k)}$ and $\mathbf{M}^{(k)}$ are finite-step Green and Martin kernels (eq. (9), eq. (10)), respectively, whose actual formulations are described in the section 4.2. These new encodings provide significant structural information while overcoming the limitations of traditional kernels, making them applicable to recurrent graphs. It can be easily checked that these SEs are invariant under graph isomorphisms.

For application to GNN models, the GKSE and MKSE are used as relative SEs in attention mechanisms or message-passing operations. Furthermore, their diagonal components are used as absolute SEs by concatenating or summing them with node features. For more details, please refer to (45).

## 4.2 Computation of finite-step Green and Martin Kernels

In this subsection, we introduce the practical method for calculating the finite-step Green and Martin kernels. While the theoretical definition of the Green and Martin kernels may seem complex, its actual computation can be efficiently performed using a recursive approach. In fact, its computation speed is comparable to that of RRWP, with the detailed computation times provided in Appendix B.2.

**Finite-step Green Kernel.** The finite-step Green kernel can be calculated using the following recurrence relation: for $k \in \mathbb{N} \cup \{0\}$,

$$\begin{cases} \mathbf{G}^{(0)} = \mathbf{I}; \\ \mathbf{G}^{(k+1)} = \mathbf{I} + \mathbf{P} \star \mathbf{G}^{(k)}, \end{cases} \tag{13}$$

where $\mathbf{I}(x, y)$ is 1 if $x = y$ and 0 otherwise. Here, $\star$ denotes convolution of kernels, which, in the case of a finite graph, corresponds to matrix multiplication.

The following theorem shows that the theoretical definition of the finite-step Green kernel (eq. (9)) and its practical computation (eq. (13)) are consistent. The proof is provided in Appendix C.2.

**Theorem 1.** *For $k \in \mathbb{N} \cup \{0\}$ and $x, y \in \mathcal{V}$, let $\mathbf{G}^{(k)}$ be the finite-step Green kernel as computed by eq. (13). Then, the following equality holds:*

$$\mathbf{G}^{(k)}(x,y) = \mathbb{E}_{\omega \in \Omega(x)}[L_y^{(k)}(\omega)], \tag{14}$$

*where $\mathbb{E}_{\omega \in \Omega(x)}[\cdot]$ means the expectation taken with respect to the probability $\mathbb{P}_{\omega \in \Omega(x)}$.*

**Finite-step Martin Kernel.** Before introducing the finite-step Martin kernel, we first observe that the traditional Martin kernel $M$ on graphs is defined by $M(x, y) := G(x, y)/G(y, y)$ for $x, y \in \mathcal{V}$. Based on this definition, the finite-step Martin kernel (with the first hitting time map) $\widetilde{\mathbf{M}}^{(k)}$ can be computed using the following formula: for $x, y \in \mathcal{V}$ and $k \in \mathbb{N} \cup \{0\}$,

$$\widetilde{\mathbf{M}}^{(k)}(x,y) = \frac{\mathbf{G}^{(k)}(x,y)}{\mathbf{G}^{(k)}(y,y)} \tag{15}$$

To apply the first return time map in the finite-step Martin kernel $\mathbf{M}^{(k)}$, we use the following modification: for $x, y \in \mathcal{V}$ and $k \in \mathbb{N} \cup \{0\}$,

$$\mathbf{M}^{(k)}(x,y) = \begin{cases} (\mathbf{P} \star \widetilde{\mathbf{M}}^{(k-1)})(x,y) & \text{if } x = y; \\ \widetilde{\mathbf{M}}^{(k)}(x,y) & \text{if } x \neq y. \end{cases} \tag{16}$$

Although the finite-step Martin kernel computed using eq. (16) may not exactly match the eq. (10), it provides a close approximation. The folowing theorem ensures that this approximation is accurate. The proof is provided in Appendix C.4.

**Theorem 2.** *For $k \in \mathbb{N} \cup \{0\}$ and $x, y \in \mathcal{V}$, let $\mathbf{M}^{(k)}$ be the finite-step Martin kernel as computed by eq. (16). Then, the following inequalities hold:*

1. $$0 \leq \mathbb{P}_{\omega \in \Omega(x)}[\tau_y^+(\omega) \leq k] - \mathbf{M}^{(k)}(x,y) \leq \frac{\mathbf{H}^{(k)}(x,y)}{\mathbf{G}^{(k-\delta(x,y))}(y,y)}; \tag{17}$$

2. $$\frac{1}{\mathbf{G}^{(k-\delta(x,y))}(y,y)} \leq \frac{\mathbf{M}^{(k)}(x,y)}{\mathbb{P}_{\omega \in \Omega(x)}[\tau_y^+(\omega) \leq k]} \leq 1, \tag{18}$$

*where $\mathbf{H}^{(k)}(x, y) = \mathbb{E}_{\omega \in \Omega(x)}[\tau_y^+(\omega) ; \tau_y^+(\omega) \leq k] = \mathbb{E}_{\omega \in \Omega(x)}[\mathbb{1}_{\{\tau_y^+(\omega) \leq k\}} \tau_y^+(\omega)]$ is the $k$-step hitting time, meaning the expectation of the first hitting time within $k$-steps and $\delta$ is the Dirac function given by $\delta(x, y) = 1$ if $x = y$ and otherwise 0.*

In the above theorem, eq. (18) ensures the approximation when $k$ is small, while eq. (17) guarantees the approximation when $k$ is large. In fact, as $k$ becomes large, $\mathbf{G}^{(k)}(y, y)$ increases sublinearly, and for finite graphs, $\mathbf{G}^{(k)}(x, y)$ converges to a specific constant. Consequently, the lower bound in eq. (18) is close to 1 when $k$ is small, and the upper bound in eq. (17) converges to 0 as $k$ becomes large.

### 4.3 EXPRESSIVENESS OF GKSE AND MKSE

As we conclude this section, we focus on the expressiveness of the newly proposed GKSE and MKSE. We begin by clarifying the concept of expressiveness comparison between two SEs. We say that $SE_1$ is more expressive than $SE_2$ if there exists continuous function that, when applied to $SE_1$, can produce $SE_2$. The following theorem compares GKSE and MKSE with other RW-based SE, specifically RRWP.

**Theorem 3.** *The following two statements hold:*

1. *The expressiveness of GKSE is equivalent to that of RRWP.*

2. *MKSE possess a unique expressiveness independent of RRWP.*

We prove the theorem in generalized RW setting, as stated in Appendix D.2. Additionally, We present further results on expressiveness in Appendix D.3 with the corresponding proofs provided in Appendix D.4.

Next, we compare GKSE and MKSE with other SE when they combined with the Generalized Distance Weisfeiler-Lehman test (GD-WL), which is a variant of Weisfeiler-Lehman test that uses a distance between nodes to update node colors as follows: for $x \in \mathcal{V}$,

$$\chi^t(x) = \text{hash}(\{\{(d(x, y), \chi^{t-1}(y)) : y \in \mathcal{V}\}\}). \tag{19}$$

The distance in GD-WL can be chosen from any graph kernel, such as the shortest path distance (SPD). By utilizing GKSE and MKSE as the distance, we obtain the following result. The proof is provided in Appendix D.4.

**Theorem 4.** *GD-WL with GKSE or MKSE is strictly stronger than GD-WL with SPD.*

## 5 EXPERIMENTAL RESULTS

### 5.1 BENCHMARKING OF GKSE AND MKSE

We evaluate GKSE and MKSE on a comprehensive suite of graph-level task benchmarks, encompassing three datasets from the Benchmarking GNNs (29) and two datasets from the Long-Range Graph Benchmark (LRGB) (46). In addition to these, we conduct experiments on the larger dataset PCQM4Mv2 from the Open Graph Benchmark - Large Scale Challenge (OGB-LSC) (30) to further validate the scalability and effectiveness of our approaches. Furthermore, we evaluate our methods on the Open Circuit Benchmark (OCB) (47), the first benchmark specifically designed for the circuit domain. Detailed descriptions of the experimental setup and configurations can be found in Appendix A.

**Benchmarking GNNs (29).** We initially test our methods on three graph-level task benchmark datasets from the BenchmarkingGNN (29): ZINC, MNIST, and CIFAR10. We primarily compare our methods against the SOTA GT model, GRIT (6), and various baselines described in Appendix A.2. To ensure a fair comparison with prior studies, we adopted experimental settings similar to those in the GraphGPS (5) and GRIT (6) papers, maintaining parameter limits of approximately 500K for ZINC and approximately 100K for MNIST and CIFAR10. Detailed hyperparameter configurations are provided in the Table 6. The experimental results are summarized in Table 1. In our experiments, GRIT+GKSE achieved SOTA performances on MNIST, and the second-best performances on ZINC and CIFAR10 when paired with GRIT+RRWP (6). GRIT+MKSE achieved SOTA performances on

Table 1: Test performance on three graph-task benchmarks from the Benchmarking GNNs (29). Shown is the mean ± s.d. of 4 runs with different random seeds. Highlighted are the top first, second, and third results.

| Model | ZINC | MNIST | CIFAR10 |
|---|---|---|---|
| | MAE↓ | Accuracy↑ | Accuracy↑ |
| GCN | 0.367 ± 0.011 | 90.705 ± 0.218 | 55.710 ± 0.381 |
| GAT | 0.384 ± 0.007 | 95.535 ± 0.205 | 64.223 ± 0.455 |
| GIN | 0.526 ± 0.051 | 96.485 ± 0.252 | 55.255 ± 1.527 |
| GraphSAGE | 0.398 ± 0.002 | 97.312 ± 0.097 | 65.767 ± 0.308 |
| GatedGCN | 0.282 ± 0.015 | 97.340 ± 0.143 | 67.312 ± 0.311 |
| PNA | 0.188 ± 0.004 | 97.940 ± 0.120 | 70.350 ± 0.630 |
| CRaW1 | 0.085 ± 0.004 | 97.944 ± 0.050 | 69.013 ± 0.259 |
| EGT | 0.108 ± 0.009 | 98.173 ± 0.087 | 68.702 ± 0.409 |
| GPS | 0.070 ± 0.004 | 98.051 ± 0.126 | 72.298 ± 0.356 |
| GRIT+RRWP | 0.059 ± 0.001 | 98.108 ± 0.111 | 76.468 ± 0.881 |
| GRIT+GKSE (ours) | 0.058 ± 0.002 | 98.305 ± 0.125 | 76.718 ± 0.919 |
| GRIT+MKSE (ours) | 0.056 ± 0.021 | 98.235 ± 0.155 | 77.365 ± 0.640 |

ZINC and CIFAR10, and exhibited the second-best performance on MNIST. These findings indicate that GKSE, and MKSE can surpass a range of existing methods on small to medium-sized datasets.

**Long-Range Graph Benchmark (46).** We further evaluate our methods on two peptide graph benchmarks from the LRGB (46) suite: Peptides-func and Peptides-struct. These benchmarks were selected to test the capability of our methods in capturing long-range dependencies within input graphs. Our methods was compared against various baselines described in Appendix A.2. Our experimental setup and hyperparameter choices closely followed those used in the baseline tested in GRIT (6), with exceptions made for batch size and RW steps. Detailed hyperparameter configurations are provided in the Table 7. The results, presented in the Table 2, indicate that on the Peptides-struct dataset, an 11-task regression benchmark, GRIT+GKSE model achieved the best performance, followed by MKSE and RRWP. On the Peptides-func dataset, a 10-label classification task, GRIT+GKSE and GRIT+MKSE performed comparably to GRIT+RRWP. These findings demonstrate our SEs' proficiency in learning long-range interactions. Notably, the superior performance of GKSE and MKSE on the Peptides-struct dataset, which uses the same graph structures as Peptides-func, suggests that our SEs are particularly effective in multi-task regression scenarios, outperforming GRIT+RRWP despite its established efficacy in multi-label classification tasks.

**PCQM4Mv2 from OGB-LSC (30).** The PCQM4Mv2 dataset (30), one of the largest molecular datasets available, serves as a critical benchmark for GTs. Our methods was compared against various baselines described in Appendix A.2. Given the extensive size of the dataset, we followed the setup of prior studies (5). Due to time constraints, we did not engage in hyperparameter exploration; instead, we utilized the hyperparameter settings from GraphGPS (5). Detailed descriptions of the experimental setup and hyperparameters can be found in Appendix A.1 and Table 7, respectively. We carried out experiments with 4 random seeds to confirm our proposed SEs' performance and found that GRIT+GKSE achieved an MAE of 0.0837, which is much better than previsouly reported results, as illustrated in the Table 3.

**Open Circuit Benchmark (47).** We conduct experiments using our methods on two datasets from the OCB (47), specifically Ckt-Bench101 and Ckt-Bench301. These datasets represent the first analog circuit benchmarks modeled as DAGs. In our evaluation, we compared our methods against various baselines described in Appendix A.2. Detailed information on dataset preparation and the experimental hyperparameters can be found in Appendix A.1 and Table 8, respectively. As shown in the left-hand side of Table 4, the GT models outperformed the MPNNs, with GRIT+GKSE achieving the best results and GRIT+MKSE achieving the second-best results or comparable with GRIT+RRWP. These results suggest that GKSE and MKSE are highly effective on datasets modeled as DAGs, further demonstrating their versatility and robustness in various graph structures.

Table 2: Test performance on two benchmarks from the LRGB (46). Shown is the mean $\pm$ s.d. of 4 runs with different random seeds. Highlighted are the top first, second, and third results.

| Model | Peptides-func | Peptides-struct |
|---|---|---|
| | AP↑ | MAE↓ |
| GCN | $0.5930 \pm 0.0023$ | $0.3496 \pm 0.0013$ |
| GINE | $0.5498 \pm 0.0079$ | $0.3547 \pm 0.0045$ |
| GatedGCN | $0.5864 \pm 0.0077$ | $0.3420 \pm 0.0013$ |
| GatedGCN+RWSE | $0.6069 \pm 0.0035$ | $0.3357 \pm 0.0006$ |
| GatedGCN+EdgeRWSE | $0.6002 \pm 0.0048$ | $0.2679 \pm 0.0015$ |
| GatedGCN+Hodge1Lap | $0.5926 \pm 0.0059$ | $0.2632 \pm 0.0008$ |
| Transformer+LapPE | $0.6326 \pm 0.0126$ | $0.2529 \pm 0.0016$ |
| SAN+LapPE | $0.6384 \pm 0.0121$ | $0.2683 \pm 0.0043$ |
| SAN+RWSE | $0.6439 \pm 0.0075$ | $0.2545 \pm 0.0012$ |
| GPS | $0.6535 \pm 0.0041$ | $0.2500 \pm 0.0005$ |
| GPS+EdgeRWSE | $0.6625 \pm 0.0042$ | $0.2501 \pm 0.0012$ |
| GPS+Hodge1Lap | $0.6584 \pm 0.0033$ | $0.2505 \pm 0.0014$ |
| GRIT+RRWP | $0.6988 \pm 0.0082$ | $0.2460 \pm 0.0012$ |
| GRIT+GKSE (ours) | $0.6976 \pm 0.0097$ | $0.2452 \pm 0.0012$ |
| GRIT+MKSE (ours) | $0.6784 \pm 0.0057$ | $0.2457 \pm 0.0013$ |

Table 3: Test performance on PCQM4Mv2 benchmark from the OGB-LSC (30). Shown is the result of a single run, excluding experiments with GKSE and MKSE, which consists of 4 runs with different random seeds. Highlighted are the top first, second, and third results.

| Model | MAE ↓ | # Param |
|---|---|---|
| GCN | 0.1379 | 2.0M |
| GCN-virtual | 0.1153 | 4.9M |
| GIN | 0.1195 | 3.8M |
| GIN-virtual | 0.1083 | 6.7M |
| TokenGT (ORF) | 0.0962 | 48.6M |
| TokenGT (Lap) | 0.0910 | 48.5M |
| GRPE | 0.0890 | 46.2M |
| EGT | 0.0869 | 89.3M |
| Graphormer | 0.0864 | 48.3M |
| Specformer-medium | 0.0916 | 4.1M |
| GPS-small | 0.0938 | 6.2M |
| GPS-medium | 0.0858 | 19.4M |
| GRIT+RRWP | 0.0859 | 16.6M |
| GRIT+GKSE (ours) | $0.0837 \pm 0.0002$ | 11.8M |
| GRIT+MKSE (ours) | $0.0839 \pm 0.0002$ | 11.8M |

## 5.2 SENSITIVITY ANALYSIS

We conducted a sensitivity analysis on the parameter $K$ for RRWP and GKSE using the Ckt-Bench101 dataset. The results, presented in the right-hand side of Table 4, provide insights into how varying $K$ impacts model performance. For this analysis, all other hyperparameters were held constant. Notably, GKSE outperformed RRWP across most values of $K$, except for $K = 15$. Furthermore, GKSE maintained the same MAE values at $K = 18$ (where RRWP performed best) even when utilizing a shorter SE length of $K = 6$. Remarkably, at an extremely short $K = 3$, the performance of GKSE remained comparable to that of GraphGPS, as indicated in the left-hand side of Table 4. These findings suggest that GKSE are highly efficient in representing DAGs. The robustness of their performance across different values of $K$ indicates their capability to effectively capture graph structures with reduced SE lengths, demonstrating their adaptability and efficiency in various graph scenarios.

Table 4: (Left) Test performance on two benchmarks from the OCB (47). Shown is the mean ± s.d. of 4 runs with different random seeds. Highlighted are the top first, second, and third results. (Right) Sensitivity Analysis of K steps of RRWP and GKSE on Ckt-Bench101 (47) dataset. Shown is the mean ± s.d. of 4 runs with different random seeds. Highlighted indicate comparable values to the GRIT+GKSE, GRIT+RRWP, and GPS+LapPE in the Ckt-Bench101 column of Left-hand side Table, respectively.

| Model | Ckt-Bench101 | Ckt-Bench301 | K | GRIT+RRWP | GRIT+GKSE |
|---|---|---|---|---|---|
| | MAE↓ | MAE↓ | | MAE↓ | MAE↓ |
| GCN | 0.0801 ± 0.0017 | 0.0584 ± 0.0006 | 3 | 0.0443 ± 0.0009 | 0.0440 ± 0.0003 |
| GAT | 0.0719 ± 0.0012 | 0.0583 ± 0.0016 | 6 | 0.0425 ± 0.0010 | 0.0418 ± 0.0010 |
| GIN | 0.0691 ± 0.0011 | 0.0528 ± 0.0004 | 9 | 0.0434 ± 0.0008 | 0.0423 ± 0.0015 |
| GraphSAGE | 0.0662 ± 0.0004 | 0.0545 ± 0.0005 | 12 | 0.0435 ± 0.0008 | 0.0429 ± 0.0010 |
| GatedGCN | 0.0668 ± 0.0006 | 0.0527 ± 0.0004 | 15 | 0.0427 ± 0.0003 | 0.0431 ± 0.0015 |
| GPS+LapPE | 0.0440 ± 0.0011 | 0.0199 ± 0.0004 | 18 | 0.0418 ± 0.0021 | 0.0395 ± 0.0033 |
| GRIT+DAGPE | 0.0444 ± 0.0011 | 0.0240 ± 0.0004 | 21 | 0.0440 ± 0.0004 | 0.0409 ± 0.0005 |
| GRIT+RRWP | 0.0418 ± 0.0021 | 0.0190 ± 0.0005 | 24 | 0.0430 ± 0.0022 | 0.0424 ± 0.0021 |
| GRIT+GKSE (ours) | 0.0395 ± 0.0033 | 0.0188 ± 0.0004 | 27 | 0.0426 ± 0.0012 | 0.0423 ± 0.0017 |
| GRIT+MKSE (ours) | 0.0409 ± 0.0016 | 0.0192 ± 0.0004 | 30 | 0.0433 ± 0.0016 | 0.0426 ± 0.0010 |

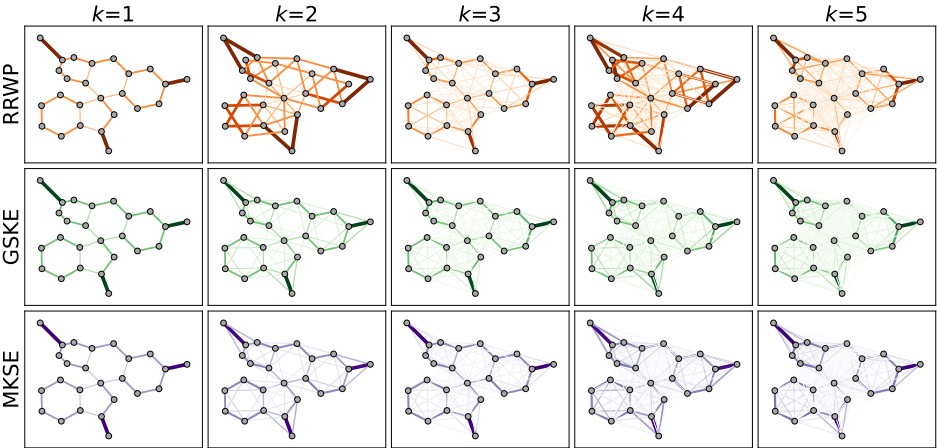

Figure 1: Visualization of RRWP, GKSE, and MKSE on a fluorescein molecule graph for $k$-steps ranging from 1 to 5. In each graph, the thickness and color intensity of the edges represent the magnitude of the corresponding SE values, with higher values indicated by thicker and darker edges.

## 5.3 Analysis of Experimental Results

Our proposed SEs exhibited SOTA on 7 out of 8 benchmarks, with superior performance on regression tasks compared to classification tasks. In particular, our proposed SEs outperformed on PCQM4Mv2 and Ckt-Bench101, which we attribute to the advantage of our proposed approaches in encapsulating graph structural information more effectively than existing SEs in certain graphs. We have explored which properties of our proposed SEs contirubte to the improvement in performance compared with another RW-based SE, specifically RRWP. We investigated the unique characteristics of molecule and circuit graphs and observed that GKSE and MKSE, compared to RRWP, represents these features in fundamentally different ways.

First, the molecular graph dataset is characterized by a large number of substructures, such as hexagonal benzene rings. In Figure 1, we visualize three SE values, RRWP, GKSE, and MKSE, on the fluorescein molecule graph. For RRWP, the edges with higher RRWP values form hexagons when $k$ is odd, on the other side they form a star shape when $k$ is even. This phenomenon arises because the hexagonal subgraph is non-aperiodic. In fact, transition probabilities on non-aperiodic graphs oscillate indefinitely. Mathematical details supporting this stability are provided in Appendix D.1. In

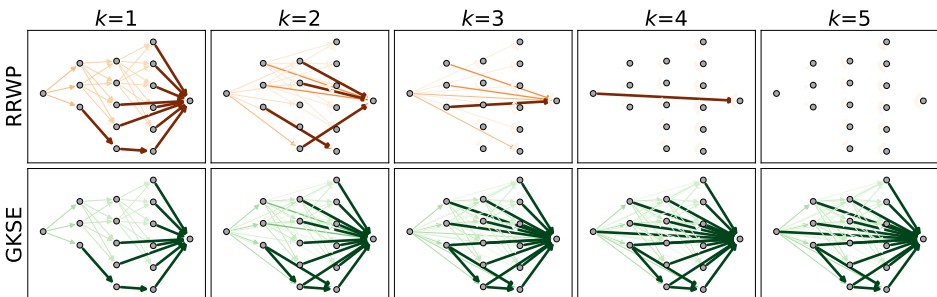

Figure 2: Visualization of RRWP and GKSE on a OCB graph sample for $k$-steps ranging from 1 to 5. In each graph, the thickness and color intensity of the edges represent the magnitude of the corresponding SE values, with higher values indicated by thicker and darker edges.

contrast, our proposed SEs provide more stable and consistent representations under non-aperiodic structures, accurately reflecting the original graph structures, as illustrated in Figure 1 and Figure 3. Based on our experimental results, we hypothesize that the stability of our proposed SEs in handling non-aperiodic substructures contributes to their improved performance. However, as our experimental results address only a limited set of cases, they do not serve as definitive evidence to confirm our assumptions. Thus, further observations and theoretical investigations are necessary to substantiate this hypothesis and gain deeper insights into the underlying mechanisms.

Second, our proposed SEs also demonstrate strong performance on datasets with DAG structures, such as OCB, which is common in the circuit domain. In DAGs, RWs terminate after a finite number of steps due to the inability to return to previously visited nodes, leading to sparse representations when using RRWP–especially for values of $K$ that are larger than diameter of the graph. This sparsity can weaken the representational power of the graph structure. Indeed, in datasets where graph samples have varying diameters, a fixed hyperparamter $K$ may fit well for some samples but result in overly sparse SEs for others with low diameters. Such imbalance can negatively affect the overall learning performance. However, our proposed SEs maintain consistent representations even for large $K$, making it suitable for capturing the structural information of DAGs, as shown in Figure 2. This enhanced efficacy on directed graphs can be attributed to the intrinsic properties of the SEs.

Overall, we infer that our proposed SEs are particularly beneficial for regression tasks involving non-aperiodic substructures or DAGs. The performance advantage of GKSE and MKSE can be attributed to their ability to effectively capture intricate structural details in such graphs, thereby enhancing the learning capabilities of GTs across diverse applications.

## 6 CONCLUSION

In this work, we introduced novel SEs, GKSE and MKSE, to expedite GTs by leveraging theoretical insights into the Green and Martin kernels within graph data. These encodings provide a foundational approach to extending RW-based methods, enhancing the expressiveness and efficiency of GTs. Our proposed SEs demonstrated significant improvements across multiple benchmarks, outperforming SOTA methods in 7 out of 8 tasks. These results confirm that our methods not only achieve superior performance but also effectively represent both molecular and circuit data, aligning with our theoretical analyses. The ability of GKSE and MKSE to capture unique structural features across diverse graph domains suggests promising directions for future research. We plan to further explore these capabilities to develop more expressive SEs with theoretically provable properties and to design model architectures that fully leverage this enhanced expressiveness. By providing a deeper understanding of the underlying kernels and a practical approach to improve GTs, this study contributes to the advancement of graph representation learning. It paves the way for developing more sophisticated and capable GT models in future research.

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

# A   EXPERIMENTAL DETAILS

## A.1   DESCRIPTION OF BENCHMARK DATASETS

A detailed overview of the statistical properties and characteristics of the benchmark datasets is presented in Table 5. The initial five datasets are sourced from the BenchmarkingGNNs (29), followed by the subsequent two from the LRGB (46), one dataset in the middle is from the OGB-LSC (30), and the final two datasets are provided by the OCB (47).

Table 5: Overview of the graph learning benchmark datasets used in this study (29; 46; 30; 47)

| Dataset | # Graphs | Avg. # nodes | Avg. # edges | Directed | Prediction level | Prediction task | Metric |
|---|---|---|---|---|---|---|---|
| ZINC | 12,000 | 23.2 | 24.9 | No | graph | regression | Mean Abs. Error |
| MNIST | 70,000 | 70.6 | 564.5 | Yes | graph | 10-class classification | Accuracy |
| CIFAR10 | 60,000 | 117.6 | 941.1 | Yes | graph | 10-class classification | Accuracy |
| PATTERN | 14,000 | 118.9 | 3,039.3 | No | inductive node | binary classification | Weighted Accuracy |
| CLUSTER | 12,000 | 117.2 | 2,150.9 | No | inductive node | 6-class classification | Weighted Accuracy |
| Peptides-func | 15,535 | 150.9 | 307.3 | No | graph | 10-task classification | Avg. Precision |
| Peptides-struct | 15,535 | 150.9 | 307.3 | No | graph | 11-task regression | Mean Abs. Error |
| PCQM4Mv2 | 3,746,620 | 14.1 | 14.6 | No | graph | regression | Mean Abs. Error |
| Ckt-Bench101 | 10,000 | 9.6 | 14.5 | Yes | graph | regression | Mean Abs. Error |
| Ckt-Bench301 | 47,248 | 9.9 | 15.5 | Yes | graph | regression | Mean Abs. Error |

**ZINC** (29) comprises 12,000 molecular graphs sampled from the ZINC database (48) of commercially available chemical compounds. These molecular graphs contain between 9 and 37 nodes, where each node corresponds to a heavy atom from one of 28 possible atom types, and each edge represents one of three possible bond types. The task associated with this dataset is to predict a molecular property known as constrained solubility (logP). The dataset is provided with a predefined split of 10,000 training, 1,000 validation, and 1,000 test samples.

**MNIST and CIFAR10** (29) are derived from their corresponding classical image classification datasets by constructing 8 nearest-neighbor graphs of SLIC superpixels for each image. The resulting graphs contain 40-75 nodes for MNIST and 85-150 nodes for CIFAR10. The 10-class classification tasks and standard dataset splits mirror the original image classification datasets, specifically 55K/5K/10K for MNIST and 45K/5K/10K for CIFAR10 in terms of train/validation/test graphs. These datasets serve as sanity checks, with most GNNs expected to achieve near 100% accuracy for MNIST and perform sufficiently well for CIFAR10.

**PATTERN and CLUSTER** (29) are synthetic datasets derived from a probabilistic block model, specifically designed for inductive node-level classification tasks. In the PATTERN dataset, the objective is to identify nodes that belong to one of 100 possible subgraph patterns. These patterns are randomly generated using different Stochastic Block Model (SBM) parameters from the rest of the graph. In the CLUSTER dataset, each graph is divided into six clusters, all generated using the same SBM distribution. Within each cluster, only one node has a unique cluster ID. The task is to determine the cluster ID for each node based on the structure of the graph.

**Peptides-func and Peptides-struct** (46) datasets consist of atomic graphs of peptides. Derived from a collection of 15,535 peptides encompassing a total of 2.3 million nodes from SATPdb (49), these two datasets share the same set of graphs but differ in their prediction tasks. In the Peptides-func dataset, the task is to classify each graph into one or more of 10 non-exclusive peptide functional classes. In the Peptides-struct dataset, the goal is to regress 11 distinct 3D structural properties of the peptides. These graphs are designed to require inference of long-range interactions (LRI) for robust performance. With an average of 150.9 nodes per graph and a mean graph diameter of 57, they provide a challenging benchmark for GTs and other GNNs aimed at capturing LRIs.

**PCQM4Mv2** (30) dataset is an extensive graph regression benchmark comprising almost 3.7 million molecular graphs. The objective is to predict the HOMO-LUMO gap, a quantum mechanical property computed using Density Functional Theory. The true labels for the original "test-dev" and "test-challenge" dataset splits are withheld by the OGB-LSC challenge organizers to ensure the integrity of the competition. Thus, we utilized the original validation set as our test set, excluding 150,000

randomly selected molecules to refine the validation process. This adjustment ensures rigorous evaluation while maintaining consistency with the dataset's intended use in benchmarking advanced GNN models.

**Ckt-Bench101 and Ckt-Bench301** (47) are pioneering datasets in the circuit domain, specifically designed for optimizing both analog circuit topologies and device parameters. Ckt-Bench101 comprises 10,000 operational amplifier (OpAmp) circuits, each topology represented as a directed acyclic graph (DAG). Ckt-Bench301 includes 47,248 OpAmp circuits, after excluding 2,752 invalid simulation results from the original 50,000 entries. For regression tasks, performance metrics for these circuits have been meticulously extracted using a circuit simulator. The OCB dataset provides critical performance metrics such as gain, bandwidth, phase margin, and a figure of merit (a composite metric of these parameters) as labels. The OCB dataset provides both subgraph-level and node-level graphs for CktGNN, a nested-GNN leveraging domain-specific knowledge of circuits. In this study, we focused on extracting node-level graph information and organizing the data for use within the GraphGPS framework (5). Each node in the dataset has node attributes of a circuit device, annotated with device-specific types and feature values, including resistance $r$, capacitance $c$, and transconductance $g_m$. Due to the lack of inherent edge attribute values in the domain, we introduced a three-dimensional edge feature vector derived from the structural properties of the graphs. These features include edge betweenness (50), edge load centrality (51; 52), and trophic differences (53), all computed using NetworkX (52). The preprocessed Ckt-Bench101 and Ckt-Bench301 datasets are provided in the supplementary materials for further research.

## A.2 BASELINES

Comparison for the BenchmarkingGNNs, we benchmark our approaches against several widely used GNN models, including prominent MPNNs and leading GNNs (GCN (54), GAT (55), GIN (14), GraphSAGE (56), GatedGCN (57), PNA (58), CRaW1 (59)); and GTs with various PE and SE ( EGT (2), GraphGPS (5), GRIT (6)). Comparison for the LRGB, we compare our methods against various MPNNs with several PESE (GCN (54), GINE (60), GatedGCN (57)) as well as several GTs (Transformer (18), SAN (3), GraphGPS (5), and GRIT (6), EdgeRWSE, and Hodge1Lap (42)). Comparison for the PCQM4Mv2, our methods was compared against two MPNNs with and without virtual nodes (GCN (54), GIN (14)) as well as several GTs (TokenGT (33), GRPE (61), EGT (2), Graphormer (1), Specformer (4), GraphGPS (5), and GRIT (6)). Comparison for the OCB, we compare our methods against various MPNNs (GCN (54), GAT (55), GIN (14), GraphSAGE (56), GatedGCN (57)) as well as two prominent GTs (GraphGPS (5), GRIT (6)). We also implemented directed acyclic graph positional encodings (DAGPE) (62) as a baseline of DAG. This comprehensive comparison ensures a robust assessment of our methods' relative performance across diverse graph benchmarks.

## A.3 DATASET SPLITS AND RANDOM SEEDS

For the datasets under evaluation, we adhere to the standard train/validation/test splits established by the benchmarks. We conduct four experimental runs on each dataset, utilizing distinct random seeds (0, 1, 2, 3). We then report both the mean performance and the standard deviation across these runs to ensure the robustness and reproducibility of our results.

## A.4 HYPERPARAMETER SETTINGS

Due to constraints in time and computational resources, an exhaustive or grid search for hyperparameters was not conducted. Instead, we primarily adhered to the hyperparameter settings of GraphGPS (5), making minor adjustments where necessary to align with commonly used parameter budgets. For benchmarking various datasets, we adhered to the standard parameter budgets widely accepted in the literature (29; 46). Specifically, we used a maximum of 500K parameters for the ZINC, PATTERN, CLUSTER, Peptides-func, and Peptides-struct datasets. For the MNIST and CIFAR10 datasets, the parameter budget was capped at 100K parameters. Across all experiments, we utilized the AdamW optimizer (63) with default settings of $\beta_1 = 0.9$, $\beta_2 = 0.999$, and $\epsilon = 10^{-8}$ same as the GraphGPS (5). The learning rate schedule featured a linear "warm-up" phase at the beginning of training, followed by a cosine decay. The duration of the warm-up period, the base

learning rate, and the total number of epochs were tuned for each dataset. The final hyperparameter configurations are detailed in Tables 6, 7, and 8.

Table 6: Hyperparameters of GKSE and MKSE for five benchmarks from the Benchmarking-GNNs (29)

| Category | Hyperparameter | ZINC | MNIST | CIFAR10 | PATTERN | CLUSTER |
|---|---|---|---|---|---|---|
| GTs | # Transformer Layers | 10 | 3 | 3 | 10 | 16 |
| | Hidden dim | 64 | 52 | 52 | 64 | 48 |
| | # Heads | 8 | 4 | 4 | 8 | 8 |
| | Dropout | 0 | 0 | 0 | 0 | 0.01 |
| | Attention dropout | 0.2 | 0.5 | 0.5 | 0.2 | 0.5 |
| | Graph pooling | sum | mean | mean | - | - |
| Training | Batch size | 32 | 16 | 8 | 32 | 16 |
| | Learning Rate | 0.001 | 0.001 | 0.001 | 0.0005 | 0.001 |
| | # Epochs | 2000 | 200 | 200 | 100 | 100 |
| | # Warmup epochs | 50 | 5 | 5 | 5 | 5 |
| | Weight decay | 1e-5 | 1e-5 | 1e-5 | 1e-5 | 1e-5 |
| GKSE | ksteps (RW-steps) | 19 | 18 | 17 | 26 | 40 |
| | PE encoder | linear | linear | linear | linear | linear |
| | # Parameters | 473,217 | 102,138 | 99,382 | 478,593 | 432,438 |
| MKSE | ksteps (RW-steps) | 18 | 16 | 17 | 14 | 41 |
| | PE encoder | linear | linear | linear | linear | linear |
| | # Parameters | 473,089 | 101,930 | 99,382 | 477,057 | 432,534 |

Table 7: Hyperparameters of GKSE and MKSE for two benchmarks from the LRGB (46) and PCQM4Mv2 benchmark from the OGB-LSC (30)

| Category | Hyperparameter | Peptides-func | Peptides-struct | PCQM4Mv2 |
|---|---|---|---|---|
| GTs | # Transformer Layers | 4 | 4 | 16 |
| | Hidden dim | 96 | 96 | 256 |
| | # Heads | 4 | 8 | 8 |
| | Dropout | 0 | 0 | 0.1 |
| | Attention dropout | 0.5 | 0.5 | 0.1 |
| | Graph pooling | mean | mean | mean |
| Training | Batch size | 8 | 32 | 256 |
| | Learning Rate | 0.0003 | 0.0003 | 0.0002 |
| | # Epochs | 200 | 200 | 150 |
| | # Warmup epochs | 5 | 5 | 10 |
| | Weight decay | 0 | 0 | 0 |
| GKSE MKSE | ksteps (RW-steps) | 26 | 24 | 16 |
| | PE encoder | linear | linear | linear |
| | # Parameters | 445,162 | 449,579 | 11.8M |

# B  SUPPLEMENTARY EXPERIMENTS

## B.1  EXPERIMENTS ON INDUCTIVE NODE-LEVEL TASK

We test our methods on two inductive node-level classification task benchmark datasets from the BenchmarkingGNN (29): PATTERN and CLUSTER. We primarily compare our methods against the SOTA GT model, GRIT (6), and various baselines described in Appendix A.2. To ensure a fair comparison with prior studies, we adopted experimental settings similar to those in the GraphGPS (5) and GRIT (6) papers, maintaining parameter limits of approximately 500K for both datasets. Detailed hyperparameter configurations are provided in the Table 6. The experimental results show that except for GRIT+GKSE performing well on the PATTERN dataset, the other results are slightly lower of comparable to the GRIT+RRWP as summarized in Table 9. We believe this is partly due to the high

Table 8: Hyperparameters of GKSE and MKSE for two benchmark datasets from the Open Circuit Benchamark (47)

| Category | Hyperparameter | Ckt-Bench101 | Ckt-Bench301 |
|---|---|---|---|
| MPNNs | # Pre Message Passing Layers | 2 | 2 |
| | # Message Passing Layers | 2 | 2 |
| | # Post Message Passing Layers | 1 | 1 |
| | Hidden dim | 64 | 64 |
| | Dropout | 0 | 0 |
| | Aggregation | mean | mean |
| GTs | # Transformer Layers | 10 | 10 |
| | Hidden dim | 64 | 64 |
| | # Heads | 8 | 8 |
| | Dropout | 0 | 0 |
| | Attention dropout | 0.2 | 0.2 |
| | Graph pooling | mean | mean |
| Training | Batch size | 32 | 64 |
| | Learning Rate | 0.001 | 0.001 |
| | # Epochs | 200 | 200 |
| | # Warmup epochs | 5 | 5 |
| | Weight decay | 1e-5 | 1e-5 |
| GPS | GPS-MPNN | GINE | GINE |
| | GPS-GlobAttn | Transformer | Transformer |
| | PE | LapPE | LapPE |
| | PE dim | 8 | 8 |
| | PE encoder | DeepSet | DeepSet |
| RRWP | ksteps (RW-steps) | 18 | 21 |
| GKSE | PE encoder | linear | linear |
| MKSE | # Parameters | 471,745 | 472,129 |

Table 9: Test performance on two inductive node-level task benchmarks from the Benchmarking GNNs. Shown is the mean $\pm$ s.d. of 4 runs with different random seeds. Highlighted are the top first, second, and third results.

| Model | PATTERN | CLUSTER |
|---|---|---|
| | W. Accuracy↑ | W. Accuracy↑ |
| GCN | $71.892 \pm 0.334$ | $68.498 \pm 0.976$ |
| GAT | $78.271 \pm 0.186$ | $70.587 \pm 0.447$ |
| GIN | $85.387 \pm 0.136$ | $64.716 \pm 1.553$ |
| GraphSAGE | $50.492 \pm 0.001$ | $63.844 \pm 0.110$ |
| GatedGCN | $85.568 \pm 0.088$ | $73.840 \pm 0.326$ |
| SAN | $86.581 \pm 0.037$ | $76.691 \pm 0.065$ |
| EGT | $86.821 \pm 0.020$ | $79.232 \pm 0.348$ |
| GPS | $86.685 \pm 0.059$ | $78.016 \pm 0.180$ |
| GRIT+RRWP | $87.196 \pm 0.076$ | $80.026 \pm 0.277$ |
| GRIT+GKSE (ours) | $87.328 \pm 0.216$ | $79.858 \pm 0.034$ |
| GRIT+MKSE (ours) | $87.150 \pm 0.194$ | $79.729 \pm 0.145$ |

average number of edges in graphs of PATTERN and CLUSTER. The dense connectivity in these datasets may hinder the ability of RWs to represent structural nuances effectively. Additionally, the

synthetic nature of the datasets, derived from SBM with specific patterns and clustring tasks, might require alternative encoding strategies better suited for dense graphs and node-level classification.

## B.2 Asymtotic Complexity, Experimental Environment and Computing Resources

The asymtotic complexities of GKSE and MKSE are $O(K|\mathcal{V}||\mathcal{E}|)$ and $O(K|\mathcal{V}||\mathcal{E}| + K|\mathcal{E}|)$ respectively, where K is the number of hops of PEs, $|\mathcal{E}|$ is the number of edges and $|\mathcal{V}|$ is the number of nodes, the asymtotic complexity of GRIT (6). We implemented our study based on the GraphGPS (5) and GRIT (6) repositories, leveraging the PyG (64) library and its GraphGym (65) module. All experiments were conducted in a compute cluster environment equipped with various CPUs, as well as NVIDIA A6000 (48GB) and A100 (40GB) GPUs. As shown in Table 10, we present the runtime and GPU memory consumption metrics for the GRIT+RRWP baseline, GKSE and MKSE on the ZINC dataset. The runtime measurements were obtained using the GraphGPS pipeline, while the GPU memory usage was monitored via the NVIDIA System Management Interface (nvidia-smi). All these experiments are carried out on a single NVIDIA A100 (40GB) GPU.

Table 10: Computing result statistics of GRIT+RRWP, GRIT+GKSE, and GRIT+MKSE on ZINC dataset with hyperparameters at Table 6

| ZINC | GRIT+RRWP | GRIT+GKSE (ours) | GRIT+MKSE (ours) |
| --- | --- | --- | --- |
| MAE↓ | $0.059 \pm 0.001$ | $0.058 \pm 0.002$ | $0.056 \pm 0.021$ |
| PE Precompute-time | 7.9 sec | 9.5 sec | 18.0 sec |
| GPU Memory | 1252MB | 1277MB | 1208MB |
| Training time | 23.8 sec/epoch | 23.6 sec/epoch | 23.1 sec/epoch |

## C Mathematical Details

In this section, we will examine the RW from the perspective of stochastic processes and discuss the specific meanings and implications of the Green kernel and Martin kernel hold in that context. For convenience, we use the following notations:

$$\mathbb{P}_x = \mathbb{P}_{\omega \in \Omega(x)}; \tag{20}$$

$$\mathbb{E}_x = \mathbb{E}_{\omega \in \Omega(x)}. \tag{21}$$

### C.1 Markov Properties

We note that a RW on a graph defined in section 3.1 is a Markov process. A Markov process possesses two key properties: the simple Markov property for fixed times and the strong Markov property for the first hitting times. These properties are stated in the following lemma, which is essential for proving various theoretical results.

For $j \in \mathbb{N} \cup \{0\}$, we define the *shift map* $\theta_j : \Omega \to \Omega$ by

$$\theta_j((\omega_0, \omega_1, \dots)) = (\omega_j, \omega_{j+1}, \dots), \quad \forall (\omega_0, \omega_1, \dots) \in \Omega. \tag{22}$$

We see that $X_i(\theta_j \omega) = X_{i+j}(\omega)$ for $i \in \mathbb{N} \cup \{0\}$ and $\omega \in \Omega$.

**Lemma 1.** *Let* $x, y \in \mathcal{V}$ *and* $j \in \mathbb{N} \cup \{0\}$. *Let* $\xi, \eta$ *be random variables with some regularity conditions. Then*

    *1. (simple Markov property)*    $\mathbb{E}_x \left[ \xi (\eta \circ \theta_j) \right] = \mathbb{E}_x \left[ \xi \, \mathbb{E}_{X_j}[\eta] \right];$

    *2. (strong Markov property)*    $\mathbb{E}_x \left[ \xi (\eta \circ \theta_{\tau_y}) \right] = \mathbb{E}_x \left[ \xi \, \mathbb{E}_y[\eta] \right].$

The detailed statement can be found in the (21). In this paper, we present only a brief version and omit the detailed conditions. Nevertheless, all random variables in the proofs below satisfy the regularity conditions.

## C.2 FINITE STEP GREEN FUNCTION

In section 4.2, we define the finite-step Green kernel $\mathbf{G}^{(k)}$ using recursive relations as in eq. (13). We can describe $\mathbf{G}^{(k)}$ in the explicit form as follows: for $k \in \mathbb{N}$ and $x, y \in \mathcal{V}$,

$$\mathbf{G}^{(k)}(x, y) = \sum_{i=0}^{k} \mathbf{P}^{(i)}(x, y), \tag{23}$$

in which $\mathbf{P}^{(0)} = \mathbf{I}$.

In order to interpret the finite-step Green kernel in terms of the stochastic process as described in the previous subsection, we first define the *counting function* $L_y^{(k)} : \Omega \to \mathbb{N} \cup \{0\}$ formally, as follows: for $\omega \in \Omega$,

$$L_y^{(k)}(\omega) = \sum_{i=0}^{k} \mathbb{1}_{\{X_i(\omega)=y\}}, \tag{24}$$

where $\mathbb{1}_{\{X_i(\cdot)=y\}} : \Omega \to \{0, 1\}$ is the *indicator function*, meaning it takes the value 1 if $X_i(\omega) = y$ and 0 otherwise. It follows directly from the definition that the value of $L_y^{(k)}(\omega)$ is equal to the number of times that the trajectory $\omega$ visits node $y$ within $k$-steps.

We now turn to Theorem 1. As described above, for $\omega \in \Omega$, the value of $L_y^{(k)}(\omega)$ is equal to the number of times that the trajectory $\omega$ visits node $y$ within $k$-steps. Recall that the probability $\mathbb{P}_x$ is concentrated on the set of all trajectories starting at node $x$. Thus, after applying the expectation $\mathbb{E}_x[\cdot]$ to the counting function $L_y^{(k)}$, its value is equal to the expected number of times that a trajectory starting at node $x$ visits node $y$ within $k$-steps. Finally, we provide the following proof at the end of this section.

*Proof of Theorem 1.* The proof is inspired by (21), which addresses the case when $k = \infty$. It follows from the definition that:

$$\mathbb{E}_x\left[L_y^{(k)}\right] = \mathbb{E}_x\left[\sum_{i=0}^{k} \mathbb{1}_{\{X_i(\omega)=y\}}\right]$$

$$= \sum_{i=0}^{k} \mathbb{E}_x\left[\mathbb{1}_{\{X_i(\omega)=y\}}\right]$$

$$= \sum_{i=0}^{k} \mathbb{P}_x[X_i(\omega) = y]$$

$$= \sum_{i=0}^{k} \mathbf{P}^i(x, y)$$

$$= \mathbf{G}^{(k)}(x, y).$$

Thus, we prove the theorem. □

## C.3 FINITE-STEP MARTIN KERNEL WITH THE FIRST HITTING TIME MAP

To prove the theorem 2, we first observe what the finite-step Martin kernel with the first hitting time map $\widetilde{\mathbf{M}}^{(k)}$ approximates. The following lemmas will be used in the proof of theorem 2.

**Lemma 2.** *For $k \in \mathbb{N} \cup \{0\}$ and $x, y \in \mathcal{V}$, let $\widetilde{\mathbf{M}}^{(k)}$ be defined in eq. (15). Then, the following equality holds:*

$$0 \le \mathbb{P}_x[\tau_y \le k] - \widetilde{\mathbf{M}}^{(k)}(x, y) \le \frac{\widetilde{\mathbf{H}}^{(k)}(x, y)}{\mathbf{G}^{(k)}(y, y)}, \tag{25}$$

*where $\widetilde{\mathbf{H}}^{(k)}(x, y) = \mathbb{E}_x[\tau_y \, ; \, \tau_y \le k] = \mathbb{E}_x[\mathbb{1}_{\{\tau_y \le k\}} \tau_y]$ is the $k$-step hitting time, meaning the expectation of the first hitting time within $k$-steps.*

*Proof.* By Theorem 1, we have

$$\mathbf{G}^{(k)}(x,y) = \mathbb{E}_x\left[L_y^{(k)}\right] = \mathbb{E}_x\left[\mathbb{1}_{\{\tau_y \le k\}} L_y^{(k)}\right]. \tag{26}$$

The second equality comes from the fact that if $\tau_y(\omega) > k$, then $L_y^{(k)}(\omega) = 0$ for all $\omega \in \Omega$.

Observe that

$$
\begin{aligned}
L_y^{(k)} &= \sum_{i=0}^{k} \mathbb{1}_{\{X_i=y\}} \\
&= \sum_{i=\tau_y}^{k} \mathbb{1}_{\{X_i=y\}} \\
&= \sum_{i=0}^{k-\tau_y} \mathbb{1}_{\{X_{i+\tau_y}=y\}} \\
&= \sum_{i=0}^{k-\tau_y} \mathbb{1}_{\{X_i=y\}} \circ \theta_{\tau_y} \\
&= L_y^{(k-\tau_y)} \circ \theta_{\tau_y}.
\end{aligned}
\tag{27}
$$

Combining with eq. (26) and eq. (27), we have

$$\mathbf{G}^{(k)}(x,y) = \mathbb{E}_x\left[\mathbb{1}_{\{\tau_y \le k\}}\left(L_y^{(k-\tau_y)} \circ \theta_{\tau_y}\right)\right] \tag{28}$$

$$= \mathbb{E}_x\left[\mathbb{1}_{\{\tau_y \le k\}}\, \mathbb{E}_y\left[L_y^{(k-\tau_y)}\right]\right] \tag{29}$$

$$= \mathbb{E}_x\left[\mathbb{1}_{\{\tau_y \le k\}}\, \mathbb{E}_y\left[L_y^{(k)}\right]\right] - \mathbb{E}_x\left[\mathbb{1}_{\{\tau_y \le k\}}\, \mathbb{E}_y\left[\sum_{i=k-\tau_y+1}^{k} \mathbb{1}_{\{X_i=y\}}\right]\right]. \tag{30}$$

Here, the strong Markov property in Lemma 1 is applied for the second equality. We note that $\mathbb{E}_y\left[L_y^{(k-\tau_y)}\right]$, which is in eq. (29), is not a constant, but a function of $\omega \in \Omega$ such that $X_0(\omega) = x$.

For the first term in eq. (30),

$$\mathbb{E}_x\left[\mathbb{1}_{\{\tau_y \le k\}}\, \mathbb{E}_y\left[L_y^{(k)}\right]\right] = \mathbb{E}_x\left[\mathbb{1}_{\{\tau_y \le k\}}\right]\mathbb{E}_y\left[L_y^{(k)}\right] = \mathbb{P}_x[\tau_y \le k]\,\mathbf{G}^{(k)}(y,y). \tag{31}$$

For the second therm in eq. (30), we first observe that for $\omega \in \Omega$ such that $X_0(\omega) = x$,

$$0 \le \sum_{i=k-\tau_y(\omega)+1}^{k} \mathbb{1}_{\{X_i(\omega)=y\}} \le \tau_y(\omega), \tag{32}$$

and hence,

$$0 \le \mathbb{E}_y\left[\sum_{i=k-\tau_y(\omega)+1}^{k} \mathbb{1}_{\{X_i(\omega)=y\}}\right] \le \mathbb{E}_y\left[\tau_y(\omega)\right] = \tau_y(\omega). \tag{33}$$

Thus the second term in eq. (30) is

$$
\begin{aligned}
0 &\le \mathbb{E}_x\left[\mathbb{1}_{\{\tau_y \le k\}}\, \mathbb{E}_y\left[\sum_{i=k-\tau_y+1}^{k} \mathbb{1}_{\{X_i=y\}}\right]\right] \\
&\le \mathbb{E}_x\left[\mathbb{1}_{\{\tau_y \le k\}}\, \tau_y\right] \\
&= \mathbb{E}_x\left[\tau_y\,;\,\tau_y \le k\right] \\
&= \widetilde{\mathbf{H}}^{(k)}(x,y).
\end{aligned}
\tag{34}
$$

Using eq. (28-34), we have

$$\mathbb{P}_x[\tau_y \leq k]\, \mathbf{G}^{(k)}(y,y) - \widehat{\mathbf{H}}^{(k)}(x,y) \leq \mathbf{G}^{(k)}(x,y) \leq \mathbb{P}_x[\tau_y \leq k]\, \mathbf{G}^{(k)}(y,y), \tag{35}$$

or

$$0 \leq \mathbb{P}_x[\tau_y \leq k] - \frac{\mathbf{G}^{(k)}(x,y)}{\mathbf{G}^{(k)}(y,y)} \leq \frac{\widehat{\mathbf{H}}^{(k)}(x,y)}{\mathbf{G}^{(k)}(y,y)}. \tag{36}$$

We complete the proof. $\qquad\square$

**Lemma 3.** *For $k \in \mathbb{N} \cup \{0\}$ and $x, y \in \mathcal{V}$, let $\widetilde{\mathbf{M}}^{(k)}$ be defined in eq. (15). Then, the following equality holds:*

$$\frac{1}{\mathbf{G}^{(k)}(y,y)} \leq \frac{\widetilde{\mathbf{M}}^{(k)}(x,y)}{\mathbb{P}_x[\tau_y \leq k]} \leq 1 \tag{37}$$

*Proof.* From eq. (29), we have

$$\mathbf{G}^{(k)}(x,y) = \mathbb{E}_x\left[\mathbb{1}_{\{\tau_y \leq k\}} \mathbb{E}_y\left[L_y^{(k-\tau_y)}\right]\right] \tag{38}$$

$$= \sum_{i=0}^{k} \mathbb{E}_x\left[\mathbb{1}_{\{\tau_y = i\}} \mathbb{E}_y\left[L_y^{(k-i)}\right]\right] \tag{39}$$

$$= \sum_{i=0}^{k} \mathbb{P}_x\left[\tau_y = i\right] \mathbf{G}^{(k-i)}(y,y) \tag{40}$$

$$\geq \sum_{i=0}^{k} \mathbb{P}_x\left[\tau_y = i\right] \tag{41}$$

$$= \mathbb{P}_x\left[\tau_y \leq k\right]. \tag{42}$$

Thus, we have

$$1 \leq \frac{\mathbf{G}^{(k)}(x,y)}{\mathbb{P}_x\left[\tau_y \leq k\right]}, \tag{43}$$

and hence

$$\frac{1}{\mathbf{G}^{(k)}(y,y)} \leq \frac{\widetilde{\mathbf{M}}^{(k)}(x,y)}{\mathbb{P}_x\left[\tau_y \leq k\right]}. \tag{44}$$

The upperbound comes from the lowerbound in Lemma 2, which is

$$0 \leq \mathbb{P}_x[\tau_y \leq k] - \widetilde{\mathbf{M}}^{(k)}(x,y). \tag{45}$$

These two inequalities complete the proof. $\qquad\square$

## C.4 Finite Step Martin Kernel

Recall that, for $y \in \mathcal{V}$, the first hitting time map $\tau_y$ and the first return time map $\tau_y^+$ is defined as follows: for $\omega \in \Omega$,

$$\tau_y(\omega) = \min\{i \in \mathbb{N} \cup \{0\} \ : \ X_i(\omega) = y\}; \tag{46}$$

$$\tau_y^+(\omega) = \min\{i \in \mathbb{N} \ : \ X_i(\omega) = y\}. \tag{47}$$

We note that if $\omega$ starts from a node other than $y$, then $\tau_y(\omega) = \tau_y^+(\omega)$. This is because random walk requires at least one step to visit another node, so the minimum in eq. (46) cannot be attained when $i = 0$.

*Proof of 1 in Theorem 2.* Let $k \in \mathbb{N}$. We first prove the case when $x = y \in \mathcal{V}$. By the definition of the first return time and the first hitting time, for $\omega \in \Omega$,

$$\begin{aligned}
\tau_x^+(\omega) &= \min\{i \geq 1 \ : \ X_i(\omega) = x\} \\
&= \min\{i \geq 0 \ : \ X_{i+1}(\omega) = x\} + 1 \\
&= \min\{i \geq 0 \ : \ X_i(\theta_1\omega) = x\} + 1 \\
&= \tau_x(\theta_1\omega) + 1.
\end{aligned} \tag{48}$$

By eq. (48), we have

$$
\begin{aligned}
\mathbb{P}_x[\tau_x^+ \le k] &= \mathbb{E}_x\left[\mathbb{1}_{\{\tau_x^+ \le k\}}\right] \\
&= \mathbb{E}_x\left[\mathbb{1}_{\{\tau_x \circ \theta_1 \le k-1\}}\right] \\
&= \mathbb{E}_x\left[\mathbb{1}_{\{\tau_x \le k-1\}} \circ \theta_1\right] \\
&= \mathbb{E}_x\left[\mathbb{E}_{X_1}\left[\mathbb{1}_{\{\tau_x \le k-1\}}\right]\right] \\
&= \mathbb{E}_x\left[\mathbb{P}_{X_1}[\tau_x \le k-1]\right] \\
&= \sum_{z \in \mathcal{V}} \mathbb{P}_x[X_1 = z]\,\mathbb{P}_z[\tau_x \le k-1] \\
&= \sum_{z \in \mathcal{V}} \mathbf{P}(x,z)\,\mathbb{P}_z[\tau_x \le k-1].
\end{aligned}
\tag{49}
$$

Here, the third equality comes from the simple Markov property in Lemma 1.

From Lemma 2,

$$
\widetilde{\mathbf{M}}^{(k-1)}(z,x) - \frac{\widetilde{\mathbf{H}}^{(k-1)}(z,x)}{\mathbf{G}^{(k-1)}(x,x)} \le \mathbb{P}_z[\tau_x \le k-1] \le \widetilde{\mathbf{M}}^{(k-1)}(z,x).
\tag{50}
$$

Combining with eq. (49) and eq. (50), we get the following two inequalities

$$
\mathbb{P}_x[\tau_x^+ \le k] \le \sum_{z \in \mathcal{V}} \mathbf{P}(x,z)\,\widetilde{\mathbf{M}}^{(k-1)}(z,x) = \left(\mathbf{P} \star \widetilde{\mathbf{M}}^{(k-1)}\right)(x,x)
\tag{51}
$$

$$
\mathbb{P}_x[\tau_x^+ \le k] \ge \sum_{z \in \mathcal{V}} \mathbf{P}(x,z)\left(\widetilde{\mathbf{M}}^{(k-1)}(z,x) - \frac{\widetilde{\mathbf{H}}^{(k-1)}(z,x)}{\mathbf{G}^{(k-1)}(x,x)}\right)
\tag{52}
$$

$$
= \left(\mathbf{P} \star \widetilde{\mathbf{M}}^{(k-1)}\right)(x,x) - \frac{\left(\mathbf{P} \star \widetilde{\mathbf{H}}^{(k-1)}\right)(x,x)}{\mathbf{G}^{(k-1)}(x,x)}
$$

By definition, $\left(\mathbf{P} \star \mathbf{M}^{(k-1)}\right)(x,x) = \mathbf{M}^{(k)}(x,x)$.

It remains to show $\left(\mathbf{P} \star \widetilde{\mathbf{H}}^{(k-1)}\right)(x,x) \le \mathbf{H}^{(k)}(x,x)$. Indeed,

$$
\begin{aligned}
\left(\mathbf{P} \star \widetilde{\mathbf{H}}^{(k-1)}\right)(x,x) &= \sum_{z \in \mathcal{V}} \mathbf{P}(x,z)\,\widetilde{\mathbf{H}}^{(k-1)}(z,x) \\
&= \sum_{z \in \mathcal{V}} \mathbb{P}_x[X_1 = z]\,\mathbb{E}_z[\tau_x \,;\, \tau_x \le k-1] \\
&= \mathbb{E}_x\left[\mathbb{E}_{X_1}\left[\mathbb{1}_{\{\tau_x \le k-1\}}\,\tau_x\right]\right] \\
&= \mathbb{E}_x\left[\left(\mathbb{1}_{\{\tau_x \le k-1\}}\,\tau_x\right) \circ \theta_1\right] \\
&= \mathbb{E}_x\left[\mathbb{1}_{\{\tau_x^+ \le k\}}(\tau_x^+ - 1)\right] \\
&= \mathbb{E}_x\left[\tau_x^+ \,;\, \tau_x^+ \le k\right] - \mathbb{P}_x[\tau_y^+ \le k] \\
&\le \mathbb{E}_x\left[\tau_x^+ \,;\, \tau_x^+ \le k\right] \\
&= \mathbf{H}^{(k)}(x,x)
\end{aligned}
\tag{53}
$$

Here, the fourth equality comes from the simple Markov property in Lemma 1 and the fifth equality comes from eq. (48).

Now, we prove the case when $x \ne y \in \mathcal{V}$. Obeserve that $\tau_y(\omega) = \tau_y^+(\omega)$ for all $\omega \in \Omega$ such that $X_0(\omega) = x$, since a RW requires at least one step to move from node $x$ to $y$. Thus we have

$$
\mathbb{P}_x[\tau_y^+ \le k] = \mathbb{P}_x[\tau_y \le k]
\tag{54}
$$

$$
\mathbf{H}^{(k)}(x,y) = \mathbb{E}_x[\tau_y^+ \,;\, \tau_y^+ \le k] = \mathbb{E}_x[\tau_y \,;\, \tau_y \le k] = \widetilde{\mathbf{H}}^{(k)}(x,y).
\tag{55}
$$

We also have $\mathbf{M}^{(k)}(x,y) = \widetilde{\mathbf{M}}^{(k)}(x,y)$ by definition. The proof follows from Lemma 2. $\qquad\square$

*Proof of 2 in Theorem 2.* Let $k \in \mathbb{N}$. We first prove the case when $x = y \in \mathcal{V}$. From Lemma 3,

$$\frac{\mathbb{P}_x[\tau_x \leq k - 1]}{\mathbf{G}^{(k-1)}(x, x)} \leq \widetilde{\mathbf{M}}^{(k-1)}(x, x) \leq \mathbb{P}_x[\tau_x \leq k - 1]. \tag{56}$$

Applying convolution to each instance with $\mathbf{P}$, by eq. (49) and the definition of $\mathbf{M}^{(k)}$, we have

$$\frac{\mathbb{P}_x[\tau_x^+ \leq k]}{\mathbf{G}^{(k-1)}(x, x)} \leq \mathbf{M}^{(k)}(x, x) \leq \mathbb{P}_x[\tau_x^+ \leq k], \tag{57}$$

which complete the proof in the case $x = y$.

Now, we prove the case when $x \neq y \in \mathcal{V}$. By eq. (54), eq. (55) and the definition of $\mathbf{M}^{(k)}$, the proof follows from the Lemma 3. $\qquad\square$

## D REPRESENTATIONAL POWER

### D.1 APERIODICITY OF GRAPHS

Let $\mathcal{G} = (\mathcal{V}, \mathcal{E})$ be a graph. The *period* $p_{\mathcal{G}}$ of $\mathcal{G}$ is defined by the greatest common divisor of the lengths of its cycles:

$$p_{\mathcal{G}} := \gcd\{n \,:\, X_0(\omega) = X_n(\omega), \, \omega \in \Omega\}. \tag{58}$$

We call a graph $\mathcal{G}$ is *aperiodic* if $p_{\mathcal{G}} = 1$ and *non-aperiodic* if $p_{\mathcal{G}} > 1$.

One important remark about the period of a graph is it affects the spectrum of the transition probability matrix $\mathbf{P}$ of the graph. By the Perron-Frobenius theorem for irreducible non-negative matrix (66), there exists exact $p_{\mathcal{G}}$ eigenvalues attaining the maximal absolute value.

For example, the hexagon graph $\mathcal{C}_6$, which is the cycle graph on 6 nodes as illustrated in Figure 3, has $p_{\mathcal{C}_6} = 2$ and its transition probability matrix has eigenvalues 1 and $-1$. In this case, the eigenvectors associated with the eigenvalues 1 and $-1$ is $\phi_1 = (1, 1, 1, 1, 1, 1)^T$ and $\phi_{-1} = (1, -1, 1, -1, 1, -1)^T$, respectively. It can be observed by spectral analysis that for any vector $\mathbf{v} \in \mathbb{R}^6$ that is not spanned by $\phi_1$ or $\phi_{-1}$, $\mathbf{P}^k \mathbf{v}$ oscillates between $\phi_1 + \phi_{-1}$ and $\phi_1 - \phi_{-1}$ as $k \to +\infty$. This phenomenon may leads to the unstability of RRWP.

Formally, let $\mathbb{1}_x$ be the one-hot vector supported at a node $x \in \mathcal{V}$. Then we have

$$\mathbf{P}(x, y) = \mathbb{1}_x^T \mathbf{P} \, \mathbb{1}_y, \quad \forall x, y \in \mathcal{V}. \tag{59}$$

Since $\mathbb{1}_y$ is not spanned by $\phi_1$ or $\phi_{-1}$, $\mathbf{P}\mathbb{1}_y$ oscillates as $k \to +\infty$ and hence $\mathbf{P}(x, y)$ also oscillates as $k \to +\infty$.

However, $\mathbf{G}^{(k)}(x, y)$ diverges to $+\infty$ and $\mathbf{M}^{(k)}(x, y)$ converges to 1 as $k \to +\infty$ for recurrent graphs. Thus, $\mathbf{G}^{(k)}$ and $\mathbf{M}^{(k)}$ do not oscillate indefinitely as illustrated in Figure 3, indicating that they are more stable under the choice of $K \in \mathbb{N} \cup \{0\}$ and capture the structural property of a graph well.

Despite the above observations, it remains unclear whether the absence of oscillation in GKSE and MKSE actually enables better detection of non-aperiodic substructures. As noted in (67), detecting specific substructures is an extremely challenging task and is proven to be infeasible with many existing GNN models. Nevertheless, since GKSE and MKSE exhibit distinct patterns compared to traditional methods, we are optimistic that they could provide some advantage. Further research is necessary to confirm this hypothesis.

### D.2 GENERALIZED RWS

Mathematically, the transition probability matrix can be defined as a real-valued matrix whose row sums equal 0 or 1. We allow the row sum to be 0 since we consider the sink node with an out-degree of 0. To avoid irregular cases, we assume that there exists a positive lower bound $\ell < 1$ for the transition probabilities. The assumption is not superflous since a transition probability matrix for a simple RW also satisfies this assumption with $\ell = 1/d_{\max}$, where $d_{\max}$ is the maximum degree of the nodes in the graph. Formally, the transition probability matrix $\mathbf{P}$ for a generalized RW satisfies

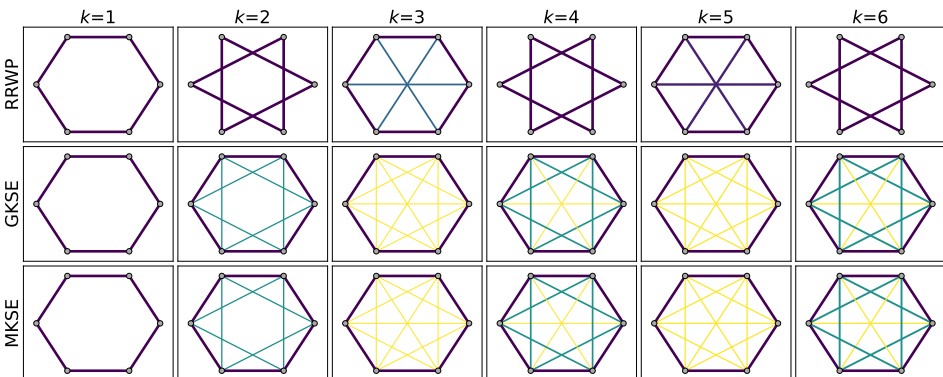

Figure 3: Visualization of RRWP, GKSE, and MKSE on a hexagon graph for $k$-steps ranging from 1 to 6.

1. $\sum_{z \in \mathcal{V}} \mathbf{P}(z, y) = 0$ or $1, \quad \forall y \in \mathcal{V}$;

2. $\mathbf{P}(x, y) > \ell, \quad \forall (x, y) \in \mathcal{E}$.

In this generalized setting, RRWP, GKSE, and MKSE can still be defined in accordance with the transition probability matrix $\mathbf{P}$ for general RWs. We will prove the theorems and corollaries in the generalized setting.

### D.3 EXPRESSIVENESS OF GKSE AND MKSE

In this section, we present several theoretical results illustrating the expressiveness of GKSE and MKSE when combined with MLP. Our findings are inspired by the study in (6), yet extend to more general scenarios involving RWs with non-identical transition probabilities. We note that, in the case of a simple RW, the transition probabilities from one node to an adjacent node in the next step are identical.

The following theorem, a restatement of Theorem 3 (1), suggests that the expressiveness of GKSE when integrated with an MLP is equivalent to that of RRWP. Analogous to the proposition in (6), we derive Corollary 1, which implies that GKSE can approximate various graph propagation matrices with precision up to an arbitrary positive error $\epsilon$. We prove the theoretical results in the general setting, specifically for non-simple RW case

**Theorem 5.** *GKSE with* MLP *has exactly the same expressive power as RRWP with* MLP.

**Corollary 1.** *Let $n, K \in \mathbb{N}$ and let $\epsilon > 0$ be sufficiently small. Then there exists* MLP *from $\mathbb{R}^K$ to $\mathbb{R}$ or $\mathbb{R}^K$ such that the for any GKSE $\in \mathbb{R}^{n \times n \times K}$ derived from a graph with $n$ nodes,* MLP(*GKSE*) *can approximate any of the following: for all $x, y \in \mathcal{V}$,*

*(a)* $\mathrm{MLP}(GKSE(x, y)) \approx \mathrm{SPD}_{K-1}(x, y)$;

*(b)* $\mathrm{MLP}(GKSE(x, y)) \approx \left( \sum_{k=0}^{K-1} \theta_k \mathbf{P}^k \right)(x, y)$;

*(c)* $\mathrm{MLP}(GKSE(x, y)) \approx (\theta_0 \mathbf{I} + \theta_1 \mathbf{A})(x, y)$

*within $\epsilon$ error. Here, $\mathrm{SPD}_{K-1}(x, y)$ represents the $K - 1$ truncated shortest path distance, and $\theta_k \in \mathbb{R}$ are arbitrary coefficients.*

We prove that MKSE possesses a unique expressiveness that cannot be achieved by RRWP alone, highlighting its potential to enhance the representational capability of GNNs in distinguishing complex graph structures. Furthermore, despite its different representational range, MKSE can also approximate several graph propagation matrices, as stated in Proposition 1. The proofs can be found in Appendix D.4. We begin by restating Theorem 3 (2) as follows.

**Theorem 6.** *RRWP with* MLP *cannot approximate MKSE.*

**Proposition 1.** *Let $n, K \in \mathbb{N}$ and let $\epsilon > 0$ be sufficiently small. Then there exists* MLP *from $\mathbb{R}^K$ to $\mathbb{R}$ or $\mathbb{R}^K$ such that the for any MKSE $\in \mathbb{R}^{n \times n \times K}$ derived from a graph with $n$ nodes and no self-loop,* MLP(*MKSE*) *can approximate any of the following: for all $x, y \in \mathcal{V}$,*

> *(a)* $\mathrm{MLP}(MKSE(x,y)) \approx \mathrm{SPD}_{K-1}(x,y)$ ;

> *(b)* $\mathrm{MLP}(MKSE(x,y)) \approx (\theta_0 \mathbf{I} + \theta_1 \mathbf{A})(x,y)$

*within $\epsilon$ error. Here, $\mathrm{SPD}_{K-1}(x,y)$ represents the $K-1$ truncated shortest path distance, and $\theta_k \in \mathbb{R}$ are arbitrary coefficients.*

## D.4 PROOFS: EXPRESSIVENESS OF GKSE AND MKSE

For convenience, we denote SEs as follows: for $K \in \mathbb{N}$,

$$(\text{RRWP}) \qquad \mathbf{R} = [\mathbf{I}, \mathbf{P}^{(1)}, \ldots, \mathbf{P}^{(K-1)}] \in \mathbb{R}^{n \times n \times K}; \tag{60}$$

$$(\text{GKSE}) \qquad \mathbf{G} = [\mathbf{I}, \mathbf{G}^{(1)}, \ldots, \mathbf{G}^{(K-1)}] \in \mathbb{R}^{n \times n \times K}; \tag{61}$$

$$(\text{MKSE}) \qquad \mathbf{M} = [\mathbf{I}, \mathbf{M}^{(1)}, \ldots, \mathbf{M}^{(K-1)}] \in \mathbb{R}^{n \times n \times K}. \tag{62}$$

*Proof of Theorem 5.* Let $K \in \mathbb{N}$. It suffices to show that there exists a continuous bijective function $\varphi : \mathbb{R}^K \to \mathbb{R}^K$ with continuous inverse such that for all $x, y \in \mathcal{V}$, $\varphi(\mathbf{G}(x,y)) = \mathbf{R}(x,y)$. The reason this completes the proof is as follows. Supppose there exists a function that can be expressed by some continuous function $f$ as $f(\mathbf{R}(x,y))$. Then, by the above observation, it is equivalent to $(f \circ \varphi)(\mathbf{G}(x,y))$. The converse also holds. Therefore, according to the standard universal approximation reuslts (68), the expressivenss of GKSE with MLP is entirely equivalent to the expressiveness of RRWP with MLP.

Now, we define the linear map $\varphi : \mathbb{R}^K \to \mathbb{R}^K$ by

$$\varphi(x_0, x_1, \ldots, x_{K-1}) = (x_0, x_1 - x_0, x_2 - x_1, \ldots, x_{K-1} - x_{K-2}). \tag{63}$$

By the definition, for all $x, y \in \mathcal{V}$, $\varphi(\mathbf{G}(x,y)) = \mathbf{R}(x,y)$. Obviously, it is continuous and has continuous inverse $\varphi^{-1}$ given by

$$\varphi^{-1}(x_0, x_1, \ldots, x_{K-1}) = (x_0, x_0 + x_1, x_0 + x_1 + x_2, \ldots, x_0 + \cdots + x_{K-1}). \tag{64}$$

This completes the proof.

$\square$

*Proof of Corollary 1.* We first prove the Proposition 3.1 from (6) in the generalized RW setting stated in Appendix D.2. Then by Theorem 5, the results follows.

We claim that for all $k = 1, \ldots, K - 1$, each nonzero entry of $\mathbf{P}^k$ is greater than $\ell^k$. We will prove the claim by using induction. The case when $k = 1$ is obvious by definition. Then we assume that the claim holds for $k$. We note that for $x, y \in \mathcal{V}$ with $\mathbf{P}^{k+1}(x,y) \neq 0$,

$$\mathbf{P}^{k+1}(x,y) = \sum_{\substack{z \in \mathcal{V}: \, \mathbf{P}^k(x,z) \neq 0 \\ \& \, (z,y) \in \mathcal{E}}} \mathbf{P}^k(x,z) \mathbf{P}(z,y) \tag{65}$$

Since $\mathbf{P}^{k+1}(x,y) \neq 0$, there exists at least one such $z \in \mathcal{V}$. Also, by assumption, $\mathbf{P}^k(x,z) > \ell^k$ and $\mathbf{P}(z,y) > \ell$. Thus we have $\mathbf{P}^{(k+1)}(x,y) > \ell^{k+1}$, proving the claim.

Following the claim, by replacing the lower bound $L$ with $\ell^{K-1}$ in the proof of Proposition 3.1 in (6), the proof is completed.

$\square$

*Proof of Theorem 6.* We will prove the theorem by providing two examples of graphs with 6 nodes for which each RRWP with MLP cannot approximate each MKSE simulteneously. Suppose that there exists a function $\varphi : \mathbb{R}^K \to \mathbb{R}^K$ constructed by MLP such that for all graphs with 6 nodes and $x, y \in \mathcal{V}$, $\varphi(\mathbf{R}(x,y))$ approximates $\mathbf{M}(x,y)$ within $\epsilon < 1/15$ error.

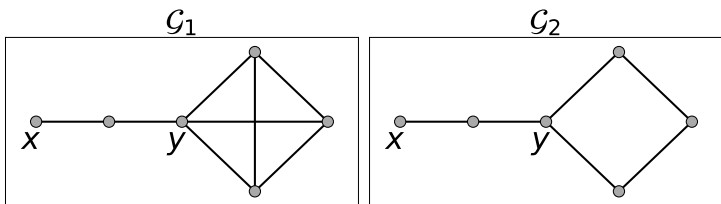

Figure 4: (Left) (4,2)-lollipop graph and (Right) A graph consisting of a 4-cycle and a 2-path connected by a single edge.

Consider the graph $\mathcal{G}_1$, which is the $(4, 2)$-lollipop graph consisting of the complete graph $\mathcal{K}_4$ on 4 nodes, the path graph $\mathcal{P}_2$ on 2 nodes, and one edge connecting $\mathcal{K}_4$ and $\mathcal{P}_2$. Also, consider the graph $\mathcal{G}_2$, which is obtained by $\mathcal{G}_1$ by replacing $\mathcal{K}_4$ with the cycle graph $\mathcal{C}_4$ on 4 nodes. Let $x, y \in \mathcal{V}$ be the nodes of $\mathcal{G}_1$ or $\mathcal{G}_2$, where $x$ is the terminal node of $\mathcal{P}_2$, and $y$ is in the $\mathcal{K}_4$ or $\mathcal{C}_4$ connected to $\mathcal{P}_2$. We visualize $\mathcal{G}_1, \mathcal{G}_2$ and $x, y$ in Figure 4.

Now, for $i = 1, 2$, we denote RRWP, GKSE, and MKSE with $K = 3$ for $\mathcal{G}_i$ by $\mathbf{R}_i$, $\mathbf{G}_i$, and $\mathbf{M}_i$, respectively. Then, we have

$$\mathbf{R}_1(x, y) = \left(0, 0, \frac{1}{2}\right)^T, \quad \mathbf{R}_2(x, y) = \left(0, 0, \frac{1}{2}\right)^T;$$

$$\mathbf{R}_1(y, y) = \left(1, 0, \frac{3}{8}\right)^T, \quad \mathbf{R}_2(y, y) = \left(1, 0, \frac{1}{2}\right)^T. \tag{66}$$

Using eq. (66), we obtain

$$\mathbf{G}_1(x, y) = \left(0, 0, \frac{1}{2}\right)^T, \quad \mathbf{G}_2(x, y) = \left(0, 0, \frac{1}{2}\right)^T;$$

$$\mathbf{G}_1(y, y) = \left(1, 1, \frac{11}{8}\right)^T, \quad \mathbf{G}_2(y, y) = \left(1, 1, \frac{3}{2}\right)^T, \tag{67}$$

and hence

$$\mathbf{M}_1(x, y) = \left(0, 0, \frac{4}{11}\right)^T, \quad \mathbf{M}_2(x, y) = \left(0, 0, \frac{1}{3}\right). \tag{68}$$

Observe that $\mathbf{R}_1(x, y) = \mathbf{R}_2(x, y)$ but $\|\mathbf{M}_1(x, y) - \mathbf{M}_2(x, y)\|_\infty = 1/15 > \epsilon$. Thus, we conclude that $\varphi(0, 0, 1/2)$ cannot approximate both $\mathbf{M}_1(x, y)$ and $\mathbf{M}_2(x, y)$ simultaneously within $\epsilon$ error. This contradiction proves the theorem.

$\square$

*Proof of Proposition 1.* From the proof of Corollary 1, for $k = 1, \ldots, K - 1$ and $x, y \in \mathcal{V}$ with $\mathbf{P}^k(x, y) \neq 0$, we have $\mathbf{P}^k(x, y) > \ell^{K-1}$.

Now, let $k = 1, \ldots, K - 1$ and let $x, y \in \mathcal{V}$ such that $\mathbf{G}^{(k)}(x, y) \neq 0$. Then, by the definition of GKSE, one of the $\mathbf{P}^i(x, y)$ is nonzero among $i = 0, \ldots k$. Thus, we have $\mathbf{G}^{(k)}(x, y) > \ell^{K-1}$. Also, we note that $\mathbf{P}^i(x, y) \leq 1$ for all $i = 1, \ldots, k$ and $x, y \in \mathcal{V}$, which implies that $\mathbf{G}^{(k)}(x, y) \leq k < K$ for all $i = 1, \ldots, k$ and $x, y \in \mathcal{V}$. Lastly, it is obvious from the definition of the MKSE that $\mathbf{G}^k(x, y) \neq 0$ iff $\mathbf{M}^{(k)}(x, y)$.

Using these observation, we have that for $k = 0, \ldots, K - 1$ and $x, y \in \mathcal{V}$ such that $\mathbf{M}^{(k)}(x, y) \neq 0$,

$$\mathbf{M}^{(k)}(x, y) = \frac{\mathbf{G}^{(k)}(x, y)}{\mathbf{G}^{(k)}(y, y)} > \frac{\ell^{K-1}}{K}. \tag{69}$$

(a) Let $f_1 : \mathbb{R}^K \to \mathbb{R}^K$ be a continuous function such that $f_1(x)_i = 0$ if $x_i \leq 0$ and 1 if $x_i \geq \ell^{K-1}/K$. Then we have that for $k = 0, \ldots, K - 1$,

$$f_1(\mathbf{M}(x, y))_k = \begin{cases} 1 & \text{if } (x \text{ can reach } y \text{ in } k \text{ hops}) \text{ or } (x = y) \\ 0 & \text{else.} \end{cases} \tag{70}$$

Let $f_2 : \mathbb{R}^K \to \mathbb{R}^K$ be defined by $f_2(x)_k = \max_{k' \leq k} x_{k'}$, which is continuous. Then we have for $k = 0, \ldots, K - 1$,

$$f_2 \circ f_1(\mathbf{M}(x, y))_k = \begin{cases} 1 & \text{if SPD}(x, y) \leq k \\ 0 & \text{else.} \end{cases} \tag{71}$$

The remainder of the proof follows the same steps as the proof of Proposition 3.1 in (6).

(b) Observe that

$$f_1(\mathbf{M}(x, y))_1 = \begin{cases} 1 & \text{if } (x \text{ can reach } y \text{ in 1 hops) or } (x = y) \\ 0 & \text{else.} \end{cases} \tag{72}$$

By the assumption that a graph have no self-loop, the cases where ($x$ can reach $y$ in 1 hops) and ($x = y$) do not occur simultaneously. Thus we have $f_1(\mathbf{M}(x, y)) = (\mathbf{I} + \mathbf{A})(x, y)$, where $\mathbf{A}$ is the adjacency matrix of the graph.

Now we take $f_3 : \mathbb{R}^K \to \mathbb{R}^2$ given by $f_3(x) = ((\theta_0 - \theta_1)x_0, \theta_1 x_1)$ and $f_4 : \mathbb{R}^2 \to \mathbb{R}$ given by $f_t(x_0, x_1) = x_0 + x_1$. Then we have

$$f_4 \circ f_3 \circ f_1(\mathbf{M}(x, y)) = \theta_0 \mathbf{I} + \theta_1 \mathbf{A}. \tag{73}$$

The remainder of the proof follows the same steps as the proof of Proposition 3.1 in (6). $\square$

*Proof of Theorem 4.* We will prove the theorem based on the proof of Proposition 3.2 in (6). We note that

$$\min\{k : \mathbf{G}^{(k)}(x, y) \neq 0\} = \min\{k : \mathbf{M}^{(k)}(x, y) \neq 0\} = \text{SPD}(x, y), \tag{74}$$

where SPD is the shortest path distance. This shows that GKSE and MKSE are more expressive than SPD, and thus, they refine SPD. Using this observation, along with Lemma 2 in (69), we conclude that GD-WL with GKSE or MKSE is stronger than GD-WL with SPD.

Next, we prove that GD-WL with GKSE or MKSE is strcitly stronger by providing some example graphs. Specifically, the Desargues graph and the Dodecahedral graph cannot be distinguished by GD-WL with SPD. However, GD-WL with GKSE or MKSE, using at least 5 steps, can distinguish between them. $\square$

