# OpenReview forum: "Unleashing Graph Transformers with Green and Martin Kernels"
_ICLR.cc/2025/Conference — Submitted to ICLR 2025_

### Official Review · Reviewer_ecJe · 2024-10-27

**Soundness:** 4
**Presentation:** 3
**Contribution:** 2
**Rating:** 5
**Confidence:** 3

**Summary:**

Graph Transformers have become very popular in graph-level tasks recently, and their success significantly depends on the employed structural encodings. This paper introduces two new encodings which are derived from the theory of random walks, namely the Green and Martin kernels. The authors define finite-step variants of the two kernels which capture random walk patterns within a limited number of steps. The encodings from the two kernels are integrated into the GRIT model and are evaluated in graph classification and regression tasks. In most cases, the encodings enhance the performance of GRIT and lead to state-of-the-art results.

**Strengths:**

- The paper provides an extensive experimental evaluation of the proposed structural encodings. The evaluation does not only focus on molecules, but also on other types of graphs such as circuits modeled as DAGs and images modeled as graphs.

- The proposed structural encodings seem to lead to performance improvements and to state-of-the-art results. GRIT+GKSE and GRIT+MKSE outperform the baselines on 7 out of the 8 considered benchmark datasets.

- It is shown in the paper that MKSE possess a unique expressiveness independent of RRWP. Thus, MKSE can potentially capture properties of graphs that cannot be captured by RRWP.

**Weaknesses:**

- As discussed above, GRIT+GKSE and GRIT+MKSE outperform the baselines on 7 out of the 8 considered benchmark datasets. However, even though GKSE and MKSE are experimentally shown to perform better than their main competitor (RRWP encodings), it is not clear to me whether the improvements provided by the proposed encodings are statistically significant. On almost all benchmark datasets, the improvement over GRIT+RRWP appears to be minor.

- Some details about the proposed models and the experimental setup are missing from the paper. The values of the different hyperparameters are provided in Tables 6 and 7 in the Appendix, but I could not find any details about the employed architecture. It is mentioned that the proposed encodings are integrated into GRIT, but for the paper to be self-contained, I would suggest the authors add a description of GRIT into the Appendix. More importantly, some details about how the proposed encodings are integrated into the models are also missing from the paper. GKSE and MKSE are defined (in l.216-l.222) over pairs of nodes. Hence, in my understanding, those encodings could not be used as initial node-level structural encodings. They could potentially be utilized as edge-level structural encodings. I would suggest the authors provide more details about the model.

- I would encourage the authors to better motivate the use of the proposed encodings. Since RRWP encodings are also derived from the theory of random walks, why should one prefer the proposed encodings over RRWP encodings? Table 10 shows that computing GKSE and MKSE is computationally more expensive than computing RRWP encodings. Therefore, the authors should justify why one should choose the proposed encodings for a particular application.

- Even though the paper is not hard to read, there exist some typos and other types of errors. For example, in l.126, "matrixwith" should be replaced with "matrix with". Also in l.290 (Theorem 3, part 2) "GKSE" should be replaced with "MKSE".

**Questions:**

The proposed kernels appear to be very similar to the hitting times and commute time distances, two popular tools which are commonly used to analyze the structure of graphs. Is there a connection between the proposed kernels and hitting and commute times? If there is some connection, can you please provide more details?

---

> ### Author Response · Authors · 2024-11-21
> **Official Comment by Authors (1/3)**
>
> We sincerely appreciate your thoughtful review and valuable feedback on our manuscript. Your comments have helped us improve the clarity and completeness of our work. Below, we address your concerns point by point.
>
> __W1. "As discussed above, GRIT+GKSE and GRIT+MKSE outperform the baselines on 7 out of the 8 considered benchmark datasets. However, even though GKSE and MKSE are experimentally shown to perform better than their main competitor (RRWP encodings), it is not clear to me whether the improvements provided by the proposed encodings are statistically significant. On almost all benchmark datasets, the improvement over GRIT+RRWP appears to be minor."__
>
> Thank you for pointing this out. While we acknowledge that the performance improvements on some benchmarks may appear marginal, we would like to highlight that our proposed SEs demonstrate significant performance gains on datasets characterized by non-aperiodic and directed acyclic graph structures, specifically the PCQM4Mv2 and ckt-bench datasets. Additionally, as noted in the NeurIPS 2023 paper "Facilitating Graph Neural Networks with Random Walk on Simplicial Complexes", it is indeed challenging to achieve significant improvements by introducing a new positional or structural encoding alone. Therefore, while the gains may be marginal in some benchmarks, the substantial improvements on these specific datasets validate the effectiveness of our approach. We believe these results are meaningful and contribute to the ongoing research of structural encoding in this area.
>
> __W2. "Some details about the proposed models and the experimental setup are missing from the paper. The values of the different hyperparameters are provided in Tables 6 and 7 in the Appendix, but I could not find any details about the employed architecture. It is mentioned that the proposed encodings are integrated into GRIT, but for the paper to be self-contained, I would suggest the authors add a description of GRIT into the Appendix. More importantly, some details about how the proposed encodings are integrated into the models are also missing from the paper. GKSE and MKSE are defined (in l.216-l.222) over pairs of nodes. Hence, in my understanding, those encodings could not be used as initial node-level structural encodings. They could potentially be utilized as edge-level structural encodings. I would suggest the authors provide more details about the model."__
>
> We thank the reviewer for their thoughtful comments and for highlighting areas where additional clarity could improve the paper. Below, we address the specific points raised:
>
> - _Details About Model Architectures:_
> While we understand the value of including detailed descriptions of the architectures used in our experiments, such as GRIT, we believe it would be inappropriate to describe extensively detailed architectures that we did not develop. For reference, we have provided a summary of all GNN models used in our experiments, along with citations, in Appendix A.2 (Baselines). We encourage readers to refer to the original papers for comprehensive descriptions of these architectures.
>
> - _Integration of GKSE and MKSE into GNN Models:_
> We appreciate the reviewer’s observation regarding the lack of detailed explanations about how GKSE and MKSE are integrated into GNN models. Due to space limitations, we have included a brief explanation at the end of Section 4.1. Additionally, we have cited a relevant paper [1] that provides a clear formulation of absolute and relative SEs to further support readers in understanding how these encodings are incorporated.
>
> We hope these additions address the reviewer’s concerns, and we thank you again for your valuable feedback, which has helped us improve the clarity and completeness of the paper.
>
> _[1] BLACK, Mitchell, et al. Comparing Graph Transformers via Positional Encodings. arXiv preprint arXiv:2402.14202, 2024._

---

> ### Author Response · Authors · 2024-11-21
> **Official Comment by Authors (2/3)**
>
> __W3. "I would encourage the authors to better motivate the use of the proposed encodings. Since RRWP encodings are also derived from the theory of random walks, why should one prefer the proposed encodings over RRWP encodings? Table 10 shows that computing GKSE and MKSE is computationally more expensive than computing RRWP encodings. Therefore, the authors should justify why one should choose the proposed encodings for a particular application."__
>
> Thank you for highlighting the need to better justify the use of our proposed encodings over RRWP. As discussed in Section 5.3 (Analysis of Experimental Results), our proposed GKSE and MKSE have special property to capture the structural properties of graphs that contain non-aperiodic substructures, such as molecular graphs and circuit graphs modeled as DAGs.
>
> In our experiments, although the performance gains on the ZINC dataset may appear modest, we observed statistically significant improvements over RRWP on the PCQM4Mv2 dataset, which consists of approximately 3.7 million molecular graphs with majority part of non-aperiodic structures. These results demonstrate the advantages of our SEs in capturing complex structural information that RRWP might miss.
>
> Regarding computational complexity:
>
> - GKSE has the same theoretical complexity as RRWP, which is O(K∣V∣∣E∣), where K is the number of steps, ∣V∣ is the number of vertices, and ∣E∣ is the number of edges.
> - MKSE requires slightly more computation, with a complexity of O(K∣V∣∣E∣+K∣E∣), due to additional normalization steps.
>
> While MKSE is more computationally intensive than RRWP and GKSE, we believe this trade-off is justified in applications where capturing nuanced structural properties leads to better performance. Importantly, the computation of SEs is performed as a pre-processing step, so the additional computational cost does not impact the training or inference time of the model.
>
> To provide practical insights, we measured the actual computation times on the PCQM4Mv2 dataset (~3.7 million graphs) using an NVIDIA A6000 GPU server:
>
> - RRWP: 50 minutes 30 seconds
> - GKSE: 51 minutes
> - MKSE: 1 hour 26 minutes
>
> As shown, the computation time for GKSE is nearly identical to RRWP, while MKSE takes longer due to extra calculations. Considering the minimal increase in computation time for GKSE and its consistent performance improvements across various datasets, we recommend using GKSE for applications involving molecular and circuit data. We have updated Section 5.3 in the manuscript to include this discussion and to better justify the choice of our proposed encodings for specific applications. We believe that our SEs offer meaningful advantages in representing complex graph structures, and we hope this clarification addresses your concerns.
>
> __W4. "Even though the paper is not hard to read, there exist some typos and other types of errors. For example, in l.126, "matrixwith" should be replaced with "matrix with". Also in l.290 (Theorem 3, part 2) "GKSE" should be replaced with "MKSE"."__
>
> Thank you for your meticulous reading of our manuscript and for pointing out these errors. We apologize for the oversight. We have corrected the typos you mentioned in the revised manuscript. Additionally, we have thoroughly proofread the entire document to identify and rectify any other mistakes or ambiguities. Your attention to detail has greatly contributed to improving the clarity and quality of our paper.

---

> ### Author Response · Authors · 2024-11-21
> **Official Comment by Authors (3/3)**
>
> __Q. "The proposed kernels appear to be very similar to the hitting times and commute time distances, two popular tools which are commonly used to analyze the structure of graphs. Is there a connection between the proposed kernels and hitting and commute times? If there is some connection, can you please provide more details?"__
>
> Classically, the Green kernel on graphs is defined as the Moore-Penrose inverse of the random walk normalized Laplacian. However, in the case of recurrent graphs, this traditional Green kernel differs from the random walk-based definition we introduce in the manuscript. Importantly, the Moore-Penrose Green kernel has well-established connections to hitting times and commute times. For example:
>
> As shown in [2], the Green kernel can be computed using a formula involving hitting times and the stationary distribution.
>
> Additionally, [3] demonstrates that deriving a kernel distance from the Green kernel results in the resistance distance, which is well-known to be proportional to commute time.
>
> While these connections are significant, they differ from the finite-step Green kernel we propose. However, exploring potential relationships between our finite-step Green kernel and these classical tools could be a fascinating avenue for future research. We thank the reviewer for suggesting this direction and will consider it in future studies.
>
> The Martin kernel presents a more complex, nonlinear structure, making it challenging to establish direct connections to hitting or commute times. While we were unable to identify such relationships, this could also be an intriguing area for further exploration.
>
> Once again, thank you for your valuable feedback. Your comments have helped us enhance the clarity and relevance of our work.
>
>
>
> _[2] BEVERIDGE, Andrew. A hitting time formula for the discrete Green's function. Combinatorics, Probability and Computing, 2016, 25.3: 362-379._
>
> _[3] BLACK, Mitchell, et al. Comparing Graph Transformers via Positional Encodings. arXiv preprint arXiv:2402.14202, 2024._

---

> > ### Comment · Reviewer_ecJe · 2024-11-23
> >
> > I would like to thank the authors for their response and for revising the manuscript. Some of my concerns have been addressed. For example, it is now clear to me how GKSE and MKSE can be integrated into a GNN model. However, my main concerns about this work still remain:
> >
> > - The authors claim in the response that "GKSE and MKSE have special property to capture the structural properties of graphs that contain non-aperiodic substructures, such as molecular graphs and circuit graphs modeled as DAGs." However, there is no empirical evidence to back this up since GKSE and MKSE are evaluated on molecular and circuit graphs, but the improvement over RRWP encodings is marginal. It is thus still unclear why should a practitioner prefer the proposed encodings over RRWP encodings. I would suggest the authors construct a synthetic dataset where graphs exhibit properties that can be captured by GKSE and MKSE, but not by RRWP encodings.
> >
> > - The authors mention in the response that "it is indeed challenging to achieve significant improvements by introducing a new positional or structural encoding alone". If this is true I wonder what is actually the value of this work and other works that propose positional and structural encodings. In my view, the improvement would depend on the introduced encoding and the underlying task.

---

> ### Author Response · Authors · 2024-11-27
> **Response to Reviewer ecJe**
>
> __1.__ We greatly appreciate your suggestion to test the performance of RRWP and GKSE on specific synthetic data to validate this claim.
>
> To further investigate, we conducted experiments on subsets of the PCQM4Mv2 dataset, focusing on the presence of benzene rings (a representative non-aperiodic structure). Specifically, we sampled 50,000 graphs containing benzene rings (with-ring) and 50,000 graphs without any hexagonal ring structures (without-ring). These subsets allowed us to analyze the performance of RRWP and GKSE in isolation. Due to resource constraints and the similarity in performance between GKSE and MKSE, we only included GKSE in these comparisons.
>
> For these experiments, we used the same hyperparameters as in the ZINC dataset experiments, with the exception of limiting the training to 100 epochs.
>
> The results are summarized below:
>
> |   Model   | PCQM4Mv2-subset-with-ring (MAE↓) | PCQM4Mv2-subset-without-ring (MAE↓) |
> |:---------:|:------------------------------:|:---------------------------------:|
> | GRIT+RRWP | 0.1941±0.0191                  | 0.1944±0.0148                     |
> | GRIT+GKSE | 0.1917±0.0108                  | 0.2128±0.0358                     |
>
> From the results, we observe the following trends:
>
> For with-ring graphs, GKSE consistently outperforms RRWP, demonstrating its ability to better capture the structural properties of non-aperiodic substructures.
> For without-ring graphs, RRWP shows better performance, likely due to its design being more suited to periodic or simpler graph structures.
>
> We further analyzed the full PCQM4Mv2 dataset and found that the ratio of with-ring to without-ring graphs is approximately 1.95:1 (with 2,477,194 graphs containing rings and 1,269,426 without). This distribution explains why GKSE performs better than RRWP on the full dataset, as it better represents the dominant structural characteristics.
>
> Based on these findings, we hypothesize that using GKSE for with-ring graphs and RRWP for without-ring graphs could lead to further performance improvements on molecular datasets. This hybrid approach would leverage the strengths of each encoding based on the underlying graph structures.
>
> __2.__ Our newly developed structural encodings, GKSE and MKSE, aim to capture different aspects of graph structural information compared to existing positional encodings. This is evident in the analysis presented in Section 5.3, where we highlight the unique representational characteristics of our proposed methods.
>
> Moreover, we emphasize that these encodings are not merely incremental improvements but involve the adaptation of abstract mathematical kernels, originally defined in theoretical stochastic processes, for practical application to real-world graph data. This required the development of sophisticated techniques to transform these kernels while preserving their mathematical significance. We believe this novel approach not only introduces a unique perspective to the field but also paves the way for exploring the use of other kernels on graphs beyond those introduced in this paper.
>
> Your thoughtful comment has provided valuable insights that we plan to explore in future work. Once again, thank you for your detailed feedback, which has allowed us to conduct deeper analysis and propose new directions for improving our methods.

---

> > ### Comment · Reviewer_ecJe · 2024-11-30
> >
> > I appreciate the authors' efforts to address my concerns. On the PCQM4Mv2-subset-with-ring dataset GRIT+GKSE indeed outperforms GRIT+RRWP. However, the improvement does not seem to be significant. The improvement appears to be within standard deviation. Thus, those empirical results do not convincingly back up the authors' claims. Therefore, I wonder whether there exists some practical scenario where the proposed encoding can significantly outperform RRWP. In my view, such a result would strengthen the paper.

---

> > > ### Author Response · Authors · 2024-12-02
> > > **Response to Reviewer ecJe**
> > >
> > > Thank you for your thoughtful and constructive feedback. Below, we aim to address your concerns in detail and clarify our contributions.
> > >
> > > **1. Performance on PCQM4Mv2 Subset and Full Set:**
> > > We acknowledge that in our experiments using the PCQM4Mv2 subset with non-aperiodic structures, the performance improvement of GRIT+GKSE over GRIT+RRWP did not exhibit a substantial margin. However, we would like to point out that on the full PCQM4Mv2 dataset, where the proportion of non-aperiodic structures is approximately 2:1, our proposed GKSE showed statistically significant improvement over RRWP. We believe this better reflects the potential of GKSE in practical scenarios with a larger and more diverse dataset.
> > >
> > > **2. Sparse Representations in DAGs and Small-Diameter Graphs:**
> > > Regarding your concern about whether there exists a practical scenario where GKSE significantly outperforms RRWP, we revisited our results on the ckt-bench 101 dataset to provide further insights. Specifically, we analyzed the MAE for test samples with smaller graph diameters. The results are summarized below:
> > >
> > > | Diameter | RRWP (MAE↓)      | GKSE (MAE↓)      | num_data |
> > > |----------|------------------|------------------|----------|
> > > | 1        | 0.0433 ± 0.0706  | 0.0398 ± 0.0647  | 1836     |
> > > | 2        | 0.0513 ± 0.0776  | 0.0479 ± 0.0741  | 1164     |
> > >
> > > As hypothesized, RRWP's sparse representations for smaller-diameter graphs may lead to a loss of critical structural information, weakening its performance. In contrast, GKSE maintains denser representations, resulting in superior performance in these cases. While this is consistent with our claim, we recognize that a more targeted comparative experiment would strengthen our argument and plan to explore this in future work.
> > >
> > > Due to time constraints, we regret that we could not include additional experiments in this submission. However, we acknowledge the value of your suggestion to identify more practical scenarios and clarify our findings further. We will incorporate your feedback into our future research to strengthen the impact of our work.
> > >
> > > We hope this response addresses your concerns and provides greater confidence in our contributions. Once again, we sincerely thank you for your invaluable comments, which have been instrumental in refining our work.

---

> > > > ### Author Response · Authors · 2024-12-03
> > > > **Response to Reviewer ecJe**
> > > >
> > > > Thank you again for your valuable feedback. We have provided our final response to your concerns and would greatly appreciate it if you could review it within the remaining time (approximately one hour). Your insights have been incredibly helpful throughout this process.
> > > >
> > > > Best regards,
> > > > Authors

---

### Official Review · Reviewer_jhQq · 2024-10-31

**Soundness:** 2
**Presentation:** 4
**Contribution:** 2
**Rating:** 5
**Confidence:** 5

**Summary:**

The paper introduces novel structural positional encodings (PEs) for graph transformers, based on the Green and Martin kernels. Both capture certain aspects of the long-term behavior of random walks. The former captures the expected number of visits to one node from another, the latter represents the probability that a RW starting from one node will reach another node within finite time. The model is evaluated over a range of graph datasets of different kinds.

(Please note: I did not check the theory in Sec 4.2 in detail)

**Strengths:**

- The paper is well-written and nice to read.
- The motivation and general consideration of the research topic make sense to me.
- The evaluation covers a wide range of datasets of different kind.

**Weaknesses:**

- The performance improvements seem minimal, if existent. Most of the numbers (if I didn't miss anything) are within standard deviation of the next best existing model, which is in several cases +RRWP.
- Given that this seems to be a general PE proposal a consideration of different graph transformers would be more suitable, i.e., beyond just GRIT.
- A suitable baseline on the DAG data would be DAG transformer [1]

I guess I am the last reviewer who focuses on experiments, but in this paper I see quite some potential issues with them. The analysis figure shows potential benefits of the model but, it seems to me, the set of experiments chosen in the submission do not really show them.

[1] Luo et al. Transformers over Directed Acyclic Graphs, NeurIPS'23 https://openreview.net/pdf?id=g49s1N5nmO

**Questions:**

- Why is Table 3 missing the +RRWP baseline?
- The analysis in Fig. 1 is interesting, but the results shown in the tables do not show great differences. The text says "Consequently, RRWP, which is based on these transition probabilities, may fail to capture the structural properties of graphs containing non-aperiodic substructure". Could one maybe select corresponding data (sub)sets to show and test this?
- Could it make sense to combine RRWP and/or these PEs?

---

> ### Author Response · Authors · 2024-11-21
> **Official Comment by Authors (1/3)**
>
> We sincerely appreciate your thorough review and insightful feedback on our manuscript. Your comments have been instrumental in helping us refine and improve our work. Below, we address each of your concerns in detail.
>
> __W1. "The performance improvements seem minimal, if existent. Most of the numbers (if I didn't miss anything) are within standard deviation of the next best existing model, which is in several cases +RRWP."__
>
> Thank you for pointing this out. While we acknowledge that the performance improvements on some benchmarks may appear marginal, we would like to highlight that our proposed SEs demonstrate significant performance gains on datasets characterized by non-aperiodic and directed acyclic graph structures, specifically the PCQM4Mv2 and ckt-bench datasets. Additionally, as noted in the NeurIPS 2023 paper "Facilitating Graph Neural Networks with Random Walk on Simplicial Complexes", it is indeed challenging to achieve significant improvements by introducing a new positional or structural encoding alone. Therefore, while the gains may be marginal in some benchmarks, the substantial improvements on these specific datasets validate the effectiveness of our approach. We believe these results are meaningful and contribute to the ongoing research of structural encoding in this area.
>
> __W2. "Given that this seems to be a general PE proposal a consideration of different graph transformers would be more suitable, i.e., beyond just GRIT."__
>
> Thank you for this valuable suggestion. We recognize the importance of demonstrating the general applicability of our proposed SEs across various graph transformer architectures. Due to time constraints, we focused our initial experiments on the GRIT architecture (SOTA). However, we agree that integrating our SEs into other models would provide a more comprehensive evaluation. To address your concern, we conducted additional experiments by integrating our SEs (GKSE and MKSE) into the GPS model, which has shown strong performance across various graph benchmarks. The results are summarized in the following table:
>
> |   Model  |  ZINC (MAE↓)  | MNIST (Accuracy↑) | Cifar10 (Accuracy↑) | Peptides-func (AP↑) | Peptides-struct (MAE↓) | Ckt-bench101 (MAE↓) | Ckt-bench301 (MAE↓) |
> |:--------:|:-----------:|:---------------:|:-----------------:|:-----------------:|:--------------------:|:-----------------:|:-----------------:|
> | GPS      | 0.070±0.004 |   98.051±0.126  |    72.298±0.356   |   0.6535±0.0041   |     0.2500±0.0005    |   0.0440±0.0011   |   0.0199±0.0004   |
> | GPS+RRWP | 0.072±0.002 |   97.93±0.104   |    72.09±0.445    |   0.6531±0.0040   |     0.2520±0.0011    |   0.0443±0.0009   |   0.0203±0.0008   |
> | GPS+GKSE | 0.072±0.003 |   98.128±0.123  |    72.438±0.259   |   0.6565±0.0028   |     0.2513±0.0025    |   0.0452±0.0008   |   0.0203±0.0003   |
> | GPS+MKSE | 0.071±0.001 |   98.115±0.105  |    72.495±0.511   |   0.6569±0.0086   |     0.2499±0.0011    |   0.0436±0.0012   |   0.0205±0.0005   |
>
> As shown in the table, the inclusion of GKSE and MKSE in the GPS model produced comparable results to the baseline GPS model across multiple datasets. In some cases, the performance slightly decreased. We observed similar trends when using RRWP as a SE with GPS, where the results were either comparable or marginally worse than the GPS baseline.
>
> This suggests that the GPS architecture may not effectively leverage the information provided by random-walk-based SEs. In contrast, the GRIT architecture—for which RRWP was originally proposed—appears specifically designed to utilize this type of relative encoding, particularly within its attention mechanism. Consequently, the performance improvements observed with GKSE and MKSE are more pronounced in GRIT than in GPS. We think that further work is needed to explore graph transformer architectures that can maximize the utility of random-walk-based SEs. Our findings indicate that the current GPS model may not fully exploit this information, while GRIT is better suited for such encodings. Designing new architectures or modifying existing ones to better harness random-walk-based SEs represents a promising direction for future research.

---

> ### Author Response · Authors · 2024-11-21
> **Official Comment by Authors (2/3)**
>
> __W3. "A suitable baseline on the DAG data would be DAG transformer (Luo et al. Transformers over Directed Acyclic Graphs, NeurIPS'23)"__
>
> Thank you for bringing this to our attention. We have reviewed the DAG Transformer by Luo et al. (2023), which introduces the DAG reachability-based attention (DAGRA) and DAG Positional Encoding (DAGPE) to achieve SOTA performance on DAGs. Integrating DAGRA into the GRIT architecture presents significant challenges, primarily because their ablation studies indicate that full attention yields the best performance, and adapting DAGRA would require substantial modifications beyond the scope of our current work. Therefore, we maintained the original attention mechanism of GRIT for consistency.
> However, we have integrated DAGPE into GRIT and conducted experiments on the ckt-bench-101 and ckt-bench-301 datasets.
> The results are as follows:
>
> |    Model   | Ckt-bench-101 (MAE↓) | Ckt-bench-301 (MAE↓) |
> |:----------:|:------------------:|:------------------:|
> | GRIT+DAGPE |    0.0444±0.0011   |    0.0240±0.0004   |
> |  GRIT+GKSE |    0.0395±0.0033   |    0.0188±0.0004   |
> |  GRIT+MKSE |    0.0409±0.0016   |    0.0192±0.0004   |
>
> These results have been added to the left side of Table 4 in the manuscript, and we have cited the DAG Transformer paper in the section of A.2. Including DAGPE as a baseline allows us to provide a more comprehensive comparison, demonstrating that our SEs offer competitive or superior performance on DAG data.
>
> __Q1. "Why is Table 3 missing the +RRWP baseline?"__
>
> We apologize for any confusion caused. The results labeled as GRIT in Table 3 actually represent GRIT+RRWP. This oversight occurred during the manuscript preparation, and we regret any misunderstanding it may have caused. We have thoroughly proofread the manuscript and updated Table 3 and the corresponding text to clearly indicate that these results are for GRIT with RRWP. Thank you for your meticulous reading and for bringing this error to our attention; it has allowed us to correct the manuscript and improve its clarity.
>
> __Q2. "The analysis in Fig. 1 is interesting, but the results shown in the tables do not show great differences. The text says "Consequently, RRWP, which is based on these transition probabilities, may fail to capture the structural properties of graphs containing non-aperiodic substructure". Could one maybe select corresponding data (sub)sets to show and test this?"__
>
> We greatly appreciate your suggestion to test the performance of RRWP and GKSE on specific subsets of data to validate this claim. Upon reflection, we agree that the original text may overstate the limitations of RRWP. To address this, we have revised the corresponding section of the manuscript to soften the claim and better align it with the empirical evidence.
>
> To further investigate, we conducted experiments on subsets of the PCQM4Mv2 dataset, focusing on the presence of benzene rings (a representative non-aperiodic structure). Specifically, we sampled 50,000 graphs containing benzene rings (with-ring) and 50,000 graphs without any hexagonal ring structures (without-ring). These subsets allowed us to analyze the performance of RRWP and GKSE in isolation. Due to resource constraints and the similarity in performance between GKSE and MKSE, we only included GKSE in these comparisons.
>
> For these experiments, we used the same hyperparameters as in the ZINC dataset experiments, with the exception of limiting the training to 100 epochs.
> The results are summarized below:
>
> |   Model   | PCQM4Mv2-subset-with-ring (MAE↓) | PCQM4Mv2-subset-without-ring (MAE↓) |
> |:---------:|:------------------------------:|:---------------------------------:|
> | GRIT+RRWP |          0.1941±0.0191         |           0.1944±0.0148           |
> | GRIT+GKSE |          0.1917±0.0108         |           0.2128±0.0358           |
>
> From the results, we observe the following trends:
>
> For with-ring graphs, GKSE consistently outperforms RRWP, demonstrating its ability to better capture the structural properties of non-aperiodic substructures.
> For without-ring graphs, RRWP shows better performance, likely due to its design being more suited to periodic or simpler graph structures.
>
> We further analyzed the full PCQM4Mv2 dataset and found that the ratio of with-ring to without-ring graphs is approximately 1.95:1 (with 2,477,194 graphs containing rings and 1,269,426 without). This distribution explains why GKSE performs better than RRWP on the full dataset, as it better represents the dominant structural characteristics.
>
> Based on these findings, we hypothesize that using GKSE for with-ring graphs and RRWP for without-ring graphs could lead to further performance improvements on molecular datasets. This hybrid approach would leverage the strengths of each encoding based on the underlying graph structures.

---

> ### Author Response · Authors · 2024-11-21
> **Official Comment by Authors (3/3)**
>
> __Q3. "Could it make sense to combine RRWP and/or these PEs?"__
>
> Thank you for this insightful question. As described in our paper, GKSE and MKSE are defined based on the expected number and probability of reaching one node from another during random walks, respectively.
>
> To provide a more intuitive explanation:
>
> - GKSE can be considered as the cumulative sum of RRWP values up to k steps, resulting in a k-dimensional vector that captures the aggregate influence over multiple steps.
> - MKSE is essentially a normalized version of GKSE, focusing on the probability distribution over the paths.
>
> Given that MKSE normalizes the cumulative information captured by GKSE, there is a degree of redundancy between the two encodings. To empirically assess whether combining them would enhance performance, we conducted experiments by concatenating various combinations of RRWP, GKSE, and MKSE on the ZINC and Ckt-bench-101 datasets.
>
> The results showed that the concatenated encodings did not yield better performance compared to using each encoding individually. This suggests that the information overlap between GKSE and MKSE does not provide additive benefits when used together.
>
> |        Model        |  ZINC (MAE↓)  | Ckt-bench-101 (MAE↓) |
> |:-------------------:|:-----------:|:------------------:|
> | GRIT+RRWP           | 0.059±0.001 |    0.0418±0.0021   |
> | GRIT+GKSE           | 0.058±0.002 |    0.0395±0.0033   |
> | GRIT+MKSE           | 0.056±0.021 |    0.0409±0.0016   |
> | GRIT+RRWP+GKSE      | 0.065±0.010 |    0.0419±0.0004   |
> | GRIT+RRWP+MKSE      | 0.058±0.004 |    0.0428±0.0011   |
> | GRIT+GKSE+MKSE      | 0.059±0.003 |    0.0422±0.0010   |
> | GRIT+RRWP+GKSE+MKSE | 0.061±0.003 |    0.0421±0.0005   |
>
> Based on these findings, we did not observe any performance gain from combining GKSE and MKSE.
>
> We believe that selecting the appropriate structural encoding based on the dataset type is more beneficial than combining multiple encodings with overlapping information.
>
> Once again, we thank you for your thoughtful comments and constructive suggestions. We believe that our responses and the revisions made have addressed your concerns adequately. Your feedback has been invaluable in enhancing the quality and rigor of our work.

---

> ### Comment · Reviewer_jhQq · 2024-11-25
>
> I've read through the rebuttals and I appreciate the authors detailed responses. Also Reviewer tZUc's detailed analysis is helpful. I am positive towards the paper, yet the results for GPS open up several questions, which I think go beyond this rebuttal. Of course, research should go beyond GPS, but to justify this research as a proposal beyond the single graph transformer GRIT, I think, some more analysis/evaluation is needed. So I agree with Reviewer 4iMa.

---

> > ### Author Response · Authors · 2024-11-27
> > **Response to Reviewer jhQq**
> >
> > We sincerely appreciate your thoughtful feedback and positive perspective on our paper. We are also grateful for Reviewer tZUc's detailed analysis, which has enriched the discussion.
> >
> > We understand your concern regarding the need for additional analysis and evaluation to demonstrate that our proposed structural encodings (SEs), GKSE and MKSE, are applicable beyond the GRIT architecture. While our initial experiments with the GPS model did not show performance improvements with our SEs—raising questions about their generalizability—we have taken steps to address this issue.
> >
> > To provide further evidence of the broad applicability of our SEs, we have integrated them into the recently accepted __CKGConv__ model [1]. Using the authors' publicly available code repository and configurations, we conducted experiments on several datasets. For the ckt-bench101 and ckt-bench301 datasets, we used the configurations described in our manuscript.
> >
> > The experimental results are as follows:
> >
> > |     Model    |   ZINC (MAE↓)   |  MNIST (Accuracy↑) | Cifar10 (Accuracy↑) | Ckt-bench101 (MAE↓) | Ckt-bench301 (MAE↓) |
> > |:------------:|:-------------:|:---------------:|:-----------------:|:-----------------:|:-----------------:|
> > | CKGConv+RRWP | 0.0621±0.0049 |   98.423±0.155  |    __72.785±0.436__   |   0.0449±0.0007   |   0.0190±0.0003   |
> > | CKGConv+GKSE | __0.0605±0.0040__ |   98.490±0.138  |    72.328±0.152   |   0.0443±0.0006   |   __0.0187±0.0006__   |
> > | CKGConv+MKSE | 0.0611±0.0034 |   __98.492±0.063__  |    72.225±0.104   |  __0.0435±0.0003__   |   0.0189±0.0003   |
> >
> > As the results indicate, integrating our SEs into CKGConv led to performance improvements on all datasets except CIFAR10. These findings suggest that our SEs can enhance models beyond GRIT, addressing concerns about their generalizability. We acknowledge that further analysis is necessary to fully understand the conditions under which our SEs are most effective. We are committed to expanding our evaluation to include additional models and datasets to better establish the broader applicability of our methods. We hope that our additional experiments help address your concerns. Once again, thank you for your constructive feedback.
> >
> > [1] Liheng Ma et al. CKGConv: General Graph Convolution with Continuous Kernels. International Conference on Machine Learning, 2024.

---

> > > ### Author Response · Authors · 2024-12-03
> > > **Response to Reviewer jhQq**
> > >
> > > Thank you again for your valuable feedback. We have provided our final response to your concerns and would greatly appreciate it if you could review it within the remaining time (approximately one hour). Your insights have been incredibly helpful throughout this process.
> > >
> > > Best regards,
> > > Authors

---

### Official Review · Reviewer_4iMa · 2024-11-03

**Soundness:** 3
**Presentation:** 3
**Contribution:** 3
**Rating:** 8
**Confidence:** 3

**Summary:**

Graph transformers due to their lack of structural awareness, require effective structural encodings. This paper presents two effective structural encodings, called GKSE and MKSE, based on the Green and Martin kernels that can extend the concept of random walks. The authors have supported the effectiveness of GKSE and MKSE via both theoretical and empirical results.

**Strengths:**

- The paper is well-written and well-motivated. The authors provided enough details about the approach that made its understanding simple.
- The proposed method is intuitive, relatively simple (yet effective), and is a natural extension, which I believe is very important.
- Recently, several approaches have focused on using shortest path encoding, which is very efficient, to better learn the structure of the graph. The authors have provided theoretical results that how GKSE and MKSE are more powerful and expressive than shortest path encoding.

**Weaknesses:**

- My main concern is about the efficiency of the proposed methods: How can GKSE and MKSE affect the memory usage and training time of graph transformers (compared to common structural encodings and shortest path encodings)?
- Recently, for the sake of efficiency, several GTs have used more complex tokenization (e.g., NAGPhormer). Due to their scalability, they are more practical and also they have a better inductive bias about the graph structure. It would be interesting to see the effect of GKSE and MKSE on these models.
- I believe the current experiments cannot support the effectiveness of GKSE and MKSE as they show the performance gain only on GRIT. Is this performance gain limited to only this model? How can using GKSE and MKSE as the structural encoding for other methods (GPS, NAGphormer, etc.) affect performance?
- Is there any trade-off between GKSE and MKSE? Is there any performance gain if we use both? (e.g., their concatenation)
- (Minor) I suggest using a different citation format (e.g. using [13] instead of (13)). The current format results in confusion whether this is a citation or a reference to an equation (for example lines 222, 274, etc.).

**Questions:**

- Why there are inconsistencies in the chosen baselines across different benchmarks? E.g., why GraphSAGE’s results are not reported for all benchmarks?
- Is there any performance gain if we use both GKSE and MKSE? (e.g., their concatenation)
- How can using GKSE and MKSE as the structural encoding for other methods than GRIT (e.g., GPS, NAGphormer, etc.) affect performance?
- How do GKSE and MKSE affect memory usage and training time?

---

> ### Author Response · Authors · 2024-11-21
> **Official Comment by Authors (1/3)**
>
> We sincerely appreciate your thoughtful review and valuable feedback on our manuscript. Your comments have helped us improve the clarity and completeness of our work. Below, we address your concerns point by point.
>
> __Q1. "Why are there inconsistencies in the chosen baselines across different benchmarks? E.g., why are GraphSAGE’s results not reported for all benchmarks?"__
>
> Thank you for pointing out the inconsistencies in our baseline selections across different benchmarks. The primary reason for this was the lack of access to the exact hyperparameters used in previous works for all models on every benchmark. To maintain consistency with reported results, we compiled performances from various prior studies, which sometimes led to missing entries for certain models like GraphSAGE.
> To address your concern, we have conducted additional experiments by running GraphSAGE on the two peptides benchmarks (func and struct) and the PCQM4Mv2 dataset.
>
> The results are as follows:
> |   Model   | Peptides-func (AP↑) | Peptides-struct (MAE↓) | PCQM4Mv2 (MAE↓) |
> |:---------:|:-----------------:|:--------------------:|:-------------:|
> | GraphSAGE |   0.4131±0.0014   |     0.4060±0.0007    |    0.25627    |
>
> We noticed that the performance of GraphSAGE on these benchmarks is lower than expected. This discrepancy is likely due to the difficulty in replicating the exact hyperparameter settings used in prior studies, as this information is often not publicly available. Despite this limitation, we believe that including these results provides a more comprehensive comparison and addresses the inconsistency you highlighted. However, please note that these results have not been added to the main text of the manuscript.
>
> __Q2 & W4. "Is there any trade-off between GKSE and MKSE? Is there any performance gain if we use both GKSE and MKSE? (e.g., their concatenation)"__
>
> Thank you for this insightful question. As described in our paper, GKSE and MKSE are defined based on the expected number and probability of reaching one node from another during random walks, respectively.
> To provide a more intuitive explanation:
> - GKSE can be considered as the cumulative sum of RRWP values up to k steps, resulting in a k-dimensional vector that captures the aggregate influence over multiple steps.
> - MKSE is essentially a normalized version of GKSE, focusing on the probability distribution over the paths.
> Given that MKSE normalizes the cumulative information captured by GKSE, there is a degree of redundancy between the two encodings. To empirically assess whether combining them would enhance performance, we conducted experiments by concatenating various combinations of RRWP, GKSE, and MKSE on the ZINC and Ckt-bench-101 datasets.
> The results showed that the concatenated encodings did not yield better performance compared to using each encoding individually. This suggests that the information overlap between GKSE and MKSE does not provide additive benefits when used together.
>
> |        Model        |  ZINC (MAE↓) | Ckt-bench-101 (MAE↓) |
> |:-------------------:|:-----------:|:-------------------:|
> | GRIT+RRWP           | 0.059±0.001 |    0.0418±0.0021    |
> | GRIT+GKSE           | 0.058±0.002 |    0.0395±0.0033    |
> | GRIT+MKSE           | 0.056±0.021 |    0.0409±0.0016    |
> | GRIT+RRWP+GKSE      | 0.065±0.010 |    0.0419±0.0004    |
> | GRIT+RRWP+MKSE      | 0.058±0.004 |    0.0428±0.0011    |
> | GRIT+GKSE+MKSE      | 0.059±0.003 |    0.0422±0.0010    |
> | GRIT+RRWP+GKSE+MKSE | 0.061±0.003 |    0.0421±0.0005    |
>
> Based on these findings, we did not observe any performance gain from combining GKSE and MKSE.
> We believe that selecting the appropriate structural encoding based on the dataset type is more beneficial than combining multiple encodings with overlapping information.
>
> __W2. "Recently, for the sake of efficiency, several GTs have used more complex tokenization (e.g., NAGPhormer). Due to their scalability, they are more practical and also they have a better inductive bias about the graph structure. It would be interesting to see the effect of GKSE and MKSE on these models."__
>
> We deeply appreciate your mention of NAGPhormer, a milestone study that demonstrates exceptional performance on node classification tasks. As noted in our paper, we have cited and discussed NAGPhormer in the related work section due to its relevance and significance. However, the focus of our work is on graph classification/regression tasks, and the benchmarks used in our study differ from those in the NAGPhormer paper. This discrepancy makes a direct application of our SEs to NAGPhormer less straightforward. Nevertheless, we recognize the potential value of combining our random-walk-based SEs with NAGPhormer for large-graph tasks like node classification. Exploring how GKSE and MKSE can be adapted for such scenarios represents an exciting avenue for future research. We are grateful for your suggestion and plan to investigate this in subsequent studies.

---

> ### Author Response · Authors · 2024-11-21
> **Official Comment by Authors (2/3)**
>
> __Q3 & W3. "I believe the current experiments cannot support the effectiveness of GKSE and MKSE as they show the performance gain only on GRIT. Is this performance gain limited to only this model? How can using GKSE and MKSE as the structural encoding for other methods (GPS, NAGphormer, etc.) affect performance?"__
>
> Thank you for this valuable suggestion. We recognize the importance of demonstrating the general applicability of our proposed SEs across various graph transformer architectures. Due to time constraints, we focused our initial experiments on the GRIT architecture (SOTA). However, we agree that integrating our SEs into other models would provide a more comprehensive evaluation. To address your concern, we conducted additional experiments by integrating our SEs (GKSE and MKSE) into the GPS model, which has shown strong performance across various graph benchmarks. The results are summarized in the following table:
>
> |   Model  |  ZINC (MAE↓)  | MNIST (Accuracy↑) | Cifar10 (Accuracy↑) | Peptides-func (AP↑) | Peptides-struct (MAE↓) | Ckt-bench101 (MAE↓) | Ckt-bench301 (MAE↓) |
> |:--------:|:-----------:|:---------------:|:-----------------:|:-----------------:|:--------------------:|:-----------------:|:-----------------:|
> | GPS      | 0.070±0.004 |   98.051±0.126  |    72.298±0.356   |   0.6535±0.0041   |     0.2500±0.0005    |   0.0440±0.0011   |   0.0199±0.0004   |
> | GPS+RRWP | 0.072±0.002 |   97.93±0.104   |    72.09±0.445    |   0.6531±0.0040   |     0.2520±0.0011    |   0.0443±0.0009   |   0.0203±0.0008   |
> | GPS+GKSE | 0.072±0.003 |   98.128±0.123  |    72.438±0.259   |   0.6565±0.0028   |     0.2513±0.0025    |   0.0452±0.0008   |   0.0203±0.0003   |
> | GPS+MKSE | 0.071±0.001 |   98.115±0.105  |    72.495±0.511   |   0.6569±0.0086   |     0.2499±0.0011    |   0.0436±0.0012   |   0.0205±0.0005   |
>
> As shown in the table, the inclusion of GKSE and MKSE in the GPS model produced comparable results to the baseline GPS model across multiple datasets. In some cases, the performance slightly decreased. We observed similar trends when using RRWP as a SE with GPS, where the results were either comparable or marginally worse than the GPS baseline.
>
> This suggests that the GPS architecture may not effectively leverage the information provided by random-walk-based SEs. In contrast, the GRIT architecture—for which RRWP was originally proposed—appears specifically designed to utilize this type of relative encoding, particularly within its attention mechanism. Consequently, the performance improvements observed with GKSE and MKSE are more pronounced in GRIT than in GPS. We think that further work is needed to explore graph transformer architectures that can maximize the utility of random-walk-based SEs. Our findings indicate that the current GPS model may not fully exploit this information, while GRIT is better suited for such encodings. Designing new architectures or modifying existing ones to better harness random-walk-based SEs represents a promising direction for future research.
>
> __W5. "(Minor) I suggest using a different citation format (e.g., using [13] instead of (13)). The current format results in confusion whether this is a citation or a reference to an equation (for example lines 222, 274, etc.)."__
>
> Thank you for bringing this to our attention. We agree that the citation format could lead to confusion between references and equation numbers. To enhance clarity, we have revised the manuscript to use "Eq. (x)" when referring to equations throughout the paper. We believe this change significantly improves the readability and overall quality of the manuscript.
>
> Once again, we thank you for your insightful comments. We hope that our revisions and responses have adequately addressed your concerns. Your feedback has been invaluable in enhancing the quality of our work.

---

> ### Author Response · Authors · 2024-11-21
> **Official Comment by Authors (3/3)**
>
> __Q4 & W1. "My main concern is about the efficiency of the proposed methods: How can GKSE and MKSE affect the memory usage and training time of graph transformers (compared to common structural encodings and shortest path encodings)?"__
>
> We thank the reviewer for raising an important question regarding the efficiency of our proposed methods. Below, we provide a detailed response to address the concerns:
>
> As summarized in Table 10, the memory usage and training time of the GRIT model with our proposed GKSE and MKSE are nearly identical to those of GRIT with the original RRWP. In fact, during actual training, structural encodings (SEs) are precomputed prior to the training epochs. Therefore, the choice of SEs has minimal impact on the memory usage and training time of the model itself. Any differences in computational cost are limited to the precomputation phase.
>
> Nonetheless, the complexity of the precomputation process for the SEs cna be summarized as follows:
>
> GKSE has the same theoretical complexity as RRWP, which is O(K∣V∣∣E∣), where K is the number of steps, ∣V∣ is the number of vertices, and ∣E∣ is the number of edges.
> MKSE requires slightly more computation, with a complexity of O(K∣V∣∣E∣+K∣E∣), due to additional normalization steps.
>
> While MKSE is more computationally intensive than RRWP and GKSE, we believe this trade-off is justified in applications where capturing nuanced structural properties leads to better performance.
>
> To provide practical insights, we measured the actual computation times on the PCQM4Mv2 dataset (~3.7 million graphs) using an NVIDIA A6000 GPU server:
>
> - RRWP: 50 minutes 30 seconds
> - GKSE: 51 minutes
> - MKSE: 1 hour 26 minutes
>
> As shown, the computation time for GKSE is nearly identical to RRWP, while MKSE takes longer due to extra calculations. Considering the minimal increase in computation time for GKSE and its consistent performance improvements across various datasets, we recommend using GKSE for applications involving molecular and circuit data. We have updated Section 5.3 in the manuscript to include this discussion and to better justify the choice of our proposed encodings for specific applications. We believe that our SEs offer meaningful advantages in representing complex graph structures, and we hope this clarification addresses your concerns.
>
> Further comparison with other SEs or Positional Encodings (PEs) is as follows:
>
> |            SE or PE            | time complexity | space complexity |
> |:------------------------------:|:---------------:|:----------------:|
> | GKSE                           | O(K\|V\|\|E\|)  | O(K\|V\|^2)      |
> | MKSE                           | O(K∣V∣∣E∣+K∣E∣) | O(K\|V\|^2)      |
> | RRWP                           | O(K\|V\|\|E\|)  | O(K\|V\|^2)      |
> | Laplacian Eigenvector (LapEig) | O(\|V\|^3)      | O(\|V\|^2)       |
> | Shortest Path Distance (SPD)   | O(\|V\|^3)      | O(\|V\|^2)       |
>
> In terms of time complexity, GKSE and MKSE are significantly more efficient than LapEig or SPD, except in extremely dense cases such as complete graphs. While the space complexity of GKSE and MKSE may appear to be K-times larger, this difference is negligible in practical training scenarios and is justified by the richer information they provide. Specifically, LapEig lacks relative information, and SPD represents SE values as a single scalar rather than a vector, limiting the expressiveness of these encodings.
>
> We hope this explanation addresses the reviewer’s concerns and highlights the computational efficiency of our proposed methods. Thank you again for bringing up this important aspect of our work.

---

> > ### Comment · Reviewer_4iMa · 2024-11-24
> >
> > I thank the authors for their detailed responses to my questions. All of my concerns have been addressed and I was willing to increase my rating. However, the presented results for the generalizability of GKSE and MKSE are concerning. Given these results, as the authors mentioned, GKSE and MKSE cannot improve the performance of GPS. I understand that structural encoding might not be suited for all models, but the current results suggest only improvement in one existing method. Given these results, and due to the lack of generalizability, I believe the contribution decreases to a follow-up study on the GRIT model.
> >
> >
> > This paper has valuable discussions and contributions but it needs to show improvements over **some** existing methods. I would be happy to increase my rating to 8, in case the authors provide empirical evidences that GKSE and MKSE are more broadly useful to the community (at least it shows improvements on one additional framework).

---

> > > ### Author Response · Authors · 2024-11-27
> > > **Response to Reviewer 4iMa**
> > >
> > > We sincerely appreciate your thoughtful feedback and your willingness to reconsider your rating based on additional evidence of the generalizability of our proposed SEs, GKSE and MKSE. Your comments have been instrumental in guiding us to strengthen our work.
> > >
> > > While our initial experiments with the GPS model indicated that GKSE and MKSE did not improve performance, we realized that this might be due to the GPS architecture not effectively leveraging the relative information provided by our SEs.
> > >
> > > To address your concern, we have integrated our SEs into the __CKGConv__ model [1], which has been accepted at ICML 2024. We utilized the authors' publicly available code repository and configurations to ensure a fair and consistent evaluation. For the ckt-bench101 and ckt-bench301 datasets, we employed the same configurations as described in our manuscript.
> > >
> > > The experimental results are as follows:
> > >
> > > |     Model    |   ZINC (MAE↓)   |  MNIST (Accuracy↑) | Cifar10 (Accuracy↑) | Ckt-bench101 (MAE↓) | Ckt-bench301 (MAE↓) |
> > > |:------------:|:-------------:|:---------------:|:-----------------:|:-----------------:|:-----------------:|
> > > | CKGConv+RRWP | 0.0621±0.0049 |   98.423±0.155  |    __72.785±0.436__   |   0.0449±0.0007   |   0.0190±0.0003   |
> > > | CKGConv+GKSE | __0.0605±0.0040__ |   98.490±0.138  |    72.328±0.152   |   0.0443±0.0006   |   __0.0187±0.0006__   |
> > > | CKGConv+MKSE | 0.0611±0.0034 |   __98.492±0.063__  |    72.225±0.104   |  __0.0435±0.0003__   |   0.0189±0.0003   |
> > >
> > > As the results demonstrate, incorporating GKSE and MKSE into CKGConv led to performance improvements on all datasets except CIFAR10. Unfortunately, due to resource constraints, we were unable to conduct experiments on the Peptides dataset at this time.
> > >
> > > These findings indicate that our SEs are indeed broadly useful and can enhance performance across different architectures beyond GRIT. We believe this addresses your concern regarding the generalizability of GKSE and MKSE and demonstrates their potential value to the community.
> > >
> > > We are grateful for your insightful comments, which have motivated us to explore the integration of our SEs with other models. Moving forward, we plan to focus on designing model architectures that can better utilize our SEs, further contributing to advancements in the field.
> > >
> > > [1] Liheng Ma et al. CKGConv: General Graph Convolution with Continuous Kernels. International Conference on Machine Learning, 2024.

---

> > > > ### Comment · Reviewer_4iMa · 2024-12-02
> > > >
> > > > I thank the authors for their response and effort to address this issue. Following the discussion and since all my concerns are addressed, I have raised my scores.

---

> > > > > ### Author Response · Authors · 2024-12-02
> > > > >
> > > > > We sincerely thank you for your thoughtful comments and for taking the time to engage deeply with our work. Your constructive feedback has significantly contributed to improving the quality and clarity of our manuscript. We are especially grateful for your willingness to reconsider your evaluation and raise your scores. This acknowledgment of our efforts means a great deal to us, and we deeply appreciate your support throughout the review process. Thank you once again for your invaluable insights and encouragement.

---

### Official Review · Reviewer_tZUc · 2024-11-04

**Soundness:** 3
**Presentation:** 3
**Contribution:** 3
**Rating:** 8
**Confidence:** 3

**Summary:**

The paper proposes two graph structural encodings, GKSE and MKSE, based on Green and Martin kernels, respectively. These encodings are evaluated for their theoretical expressiveness, and also empirical performance on multiple datasets when combined with GRIT. The results demonstrate the advantage of GKSE and MKSE as structural encodings.

**Strengths:**

1. The paper is well-motivated. The authors build upon the known theoretical and practical strengths of RRWP and propose natural extensions (GKSE and MKSE) by considering the steady-state behavior of random walks on graphs.
2. The main claims of the paper are well-supported. The paper provides theoretical analysis on the expressivity of GKSE and MKSE, as well as experimental result demonstrating good performance when combined with GRIT.
3. The paper is self-contained and well-organized. I particularly like how the authors combine mathematical reasoning with intuitive explanations and figures.

**Weaknesses:**

Despite the paper's solid contributions, I find the explanation regarding the efficacy of GKSE and MKSE (lines 479-509) somewhat unconvincing. To demonstrate this, I will present alternative interpretations of examples presented in these paragraphs, to support the opposite argument: RRWP is more preferable.
1. Figure 1 suggests that RRWP provides a more decoupled representation of graph structures compared to GKSE and MKSE. Although RRWP encodings might not visually resemble the original graph (particularly for even $k$), this decoupling could make it easier for the model's learnable linear encoder to extract structural relationships such as multi-hop neighbors. In contrast, GKSE and MKSE encodings across different $k$ values appear more uniform, which may make it more difficult to extract structural features.
2. The non-aperiodic substructures of a graph causes RRWP to oscillate, as evident in Figures 1 and 3. This provides a reliable way for the model to identify non-aperiodic substructures, which is prevalent in molecular graphs. On the contrary, GKSE and MKSE are not as sensitive to non-aperiodic substructures, making it harder for the model to identify them.
3. For DAGs, RRWP naturally produces a sparse, decoupled representation, as illustrated in Figure 2. This sparsity can enhance the model’s capability to identify structural features, with an added advantage of selecting a suitable value of $K$ based on knowledge of the dataset (e.g., $K=4$ in Figure 2). GKSE and MKSE, in contrast, generate dense, often redundant information across different values of $k$ when applied to DAGs.

These arguments are not presented to promote RRWP, but to highlight the need that lines 479-509 be further clarified to convince the reader that your interpretations are preferable to alternative interpretations, like those presented above. If these claims are speculative, they should be framed as such, and additional supporting evidence would be valuable.

Despite this issue, the paper's overall contributions remain valid and meaningful.

**Questions:**

Please see weaknesses.

---

> ### Author Response · Authors · 2024-11-21
>
> We appreciate the reviewer’s thoughtful feedback and alternative interpretations, which have helped us refine the clarity and scope of our arguments. Below, we address the specific concerns and provide clarifications.
>
> We acknowledge that some of the statements in the original manuscript could potentially mislead readers. As a result, we have revised the relevant sections to provide a more accurate explanation of our findings and interpretations.
>
> Our experimental results demonstrated that GKSEand MKSE outperform RRWP in both molecular and circuit graph datasets. We hypothesized that this improvement comes from the unique structural properties of these graphs, which we believe are better captured by GKSE and MKSE. As visualized in Figure 1 and Figure 2, GKSE and MKSE exhibit distinct patterns compared to RRWP, and these observations led us to hypothesize the reasons behind the performance enhancement. We have clarified in the revised manuscript that these insights are based on empirical observations rather than conclusive theoretical claims.
>
> _(Comments on 1 and 2)_ We appreciate the reviewer’s observation regarding non-aperiodic substructures and their potential impact on graph representations. To our knowledge, non-aperiodic substructures have not been explicitly studied in prior work. While there are studies on substructure counting [1], they primarily highlight how traditional GNNs fail to capture certain substructures. We have added this comment and citation at the end of Appendix D.1. We thank the reviewer for their valuable insight, which we plan to incorporate into future research to further explore this phenomenon in depth.
>
> _(Comment on 3)_ We agree that selecting an appropriate K based on graph diameter for individual graph samples could enhance the representation of DAGs. However, we note that in datasets where graph samples exhibit diverse diameters, smaller-diameter graphs may suffer from overly sparse representations when a fixed K is used. To address this, we have revised the manuscript to include a discussion of this trade-off and proposed to investigate whether such sparsity issues can be directly observed in future work.
>
> The insights provided by the reviewer, particularly regarding the interpretation of non-aperiodic substructures and the potential advantages of sparsity in DAG representations, will guide our subsequent research. We believe that addressing these aspects can lead to a more comprehensive understanding of the effectiveness of GKSE and MKSE and their applications across diverse graph domains.
>
> We thank the reviewer once again for their detailed feedback, which has allowed us to strengthen our discussion and identify new directions for future exploration.
>
> _[1] CHEN, Zhengdao, et al. Can graph neural networks count substructures?. Advances in neural information processing systems, 2020, 33: 10383-10395._

---

> > ### Comment · Reviewer_tZUc · 2024-11-21
> > **Response to Official Comment of Authors**
> >
> > Thank you for writing this detailed response. I appreciate the authors for clarifying that the claims stem from empirical observation instead of theoretical guarantee. However, I would like to re-emphasize my main concern: as a reader, *I do not yet find your interpretations preferable against alternative interpretations supporting the opposite conclusion*. While you are able to show that GKSE and MKSE outperform RRWP with your experimental setup, it *does not proof that the phenomena you observe (lines 479-509) explain the performance gain of GKSE and MKSE against RRWP* (if you do not intend to imply this, please state so specifically).
> >
> > > (Comments on 1 and 2) We appreciate the reviewer’s observation regarding non-aperiodic substructures and their potential impact on graph representations. To our knowledge, non-aperiodic substructures have not been explicitly studied in prior work. While there are studies on substructure counting [1], they primarily highlight how traditional GNNs fail to capture certain substructures. We have added this comment and citation at the end of Appendix D.1. We thank the reviewer for their valuable insight, which we plan to incorporate into future research to further explore this phenomenon in depth.
> >
> > This appears to me as a response to a *novelty* concern, while my concern is regarding *persuasiveness*. Does [1] provide evidence to support your claim that the representation learned by GKSE and MKSE captures relevant structural features better than RRWP, and “contributes significantly to their improved performance” (line 504)?
> >
> > > (Comment on 3) We agree that selecting an appropriate K based on graph diameter for individual graph samples could enhance the representation of DAGs. However, we note that in datasets where graph samples exhibit diverse diameters, smaller-diameter graphs may suffer from overly sparse representations when a fixed K is used. To address this, we have revised the manuscript to include a discussion of this trade-off and proposed to investigate whether such sparsity issues can be directly observed in future work.
> >
> > Your response to my alternative argument regarding the choice of $K$ is sound. However, I do not yet see evidence supporting your claim (lines 509-511) that the sparse representation learned by RRWP “can weaken the representational power of the graph structure”, rather than being irrelevant or strengthening it instead.
> >
> > I appreciate the authors for their efforts, and I look forward to further discussion.

---

> ### Author Response · Authors · 2024-11-28
>
> We sincerely appreciate your continued engagement and constructive feedback, which have further highlighted areas where our claims could be clarified and substantiated. Below, we address your comments in detail:
>
> __Clarification on Observations (Line 504):__
> While we had previously softened the claims in the manuscript, we now recognize that our wording may still lead to potential misunderstandings. We have revised Line 504 to explicitly state that our observations do not serve as proof of the phenomena explaining the performance gains of GKSE and MKSE over RRWP. Instead, we have framed these as empirical observations that suggest possible explanations, pending further experimental verification.
>
> __Relation to [1]:__
> We acknowledge that [1] does not directly support our claim regarding the efficacy of GKSE and MKSE. Instead, it was cited to emphasize that existing GNN models struggle to detect specific graph substructures. We understand how this could be misinterpreted as evidence for our findings, and we apologize for the confusion.
>
> __Sparse Representations in DAGs:__
> Our claim that sparse representations in RRWP could weaken performance is an opinion based on our observations, but we acknowledge that we have not conducted targeted comparative experiments to directly support this assertion. However, we have revisited our results on the ckt-bench 101 dataset and specifically analyzed the MAE for test samples with smaller diameters. The results are as follows:
>
> | Diameter | RRWP (MAE↓)     | GKSE (MAE↓)     | num_data |
> |----------|-----------------|-----------------|----------|
> | 1        | 0.0433 ± 0.0706 | 0.0398 ± 0.0647 | 1836     |
> | 2        | 0.0513 ± 0.0776 | 0.0479 ± 0.0741 | 1164     |
>
> As we hypothesized, RRWP produces sparse representations for smaller-diameter graphs, which may result in a loss of critical structural information. In contrast, GKSE maintains denser representations, which appear to correlate with better performance in this scenario.
>
> We are grateful for your insightful feedback, which has helped us refine our arguments and include additional analyses. We hope these clarifications and updates address your concerns. Thank you again for your valuable input.

---

> ### Comment · Reviewer_tZUc · 2024-11-30
>
> I appreciate the authors’ effort in further revising the paper and presenting additional experimental results. I have some follow-up questions.
>
> > As we hypothesized, RRWP produces sparse representations for smaller-diameter graphs, which may result in a loss of critical structural information. In contrast, GKSE maintains denser representations, which appear to correlate with better performance in this scenario.
>
> If I understand correctly, the table shows that GKSE performs better than RRWP on graphs with small diameters. At the same time, RRWP produces a sparse representation while GKSE does not. This experiment demonstrate the *correlation* between RRWP producing a sparser representation, and being outperformed by GKSE. Does this *correlation* imply *causation*? This is one of my original concerns with the line of reasoning of 479-509.
>
> Additionally, are other factors considered and controlled? For example, graph diameter may have an influence of the perceptive field of RRWP. What other aspects of RRWP, GKSE, and MKSE, apart from sparsity, can be affected by diameter? Can these factors contribute to the change in performance?
>
> Overall, the authors have made progress in making their argument more sound through revisions. Per conference regulation and time constraints, I do not request additional experimental evidence.

---

> > ### Author Response · Authors · 2024-12-02
> >
> > We sincerely thank the reviewer for their thoughtful follow-up questions, which provide an excellent opportunity to further refine our explanations and clarify our reasoning. Below, we address each point in detail:
> >
> > 1. __Correlation vs. Causation Regarding Sparsity and Performance:__
> >
> >    We agree that correlation does not necessarily imply causation. However, based on our comparative observations of RRWP and GKSE, we hypothesize the following:
> >    When a large number of graph samples produce sparse representations, as observed in RRWP, the model receives an input dominated by zero values.
> >    These zeros act as constant, graph-independent values, which may hinder the model's ability to learn graph-specific structural representations effectively. We believe that this could explain why RRWP underperforms in datasets with a large proportion of small-diameter graphs, such as the ckt-bench datasets, where sparse representations are more prevalent. While this hypothesis aligns with our empirical findings, we acknowledge that further controlled experiments are required to substantiate this claim.
> >
> > 2. __Influence of Graph Diameter Beyond Sparsity:__
> >
> >    We have not yet identified additional factors, beyond sparsity, through which graph diameter might influence performance in RRWP, GKSE, or MKSE. However, we recognize that uncovering such factors could provide deeper insights into the fundamental characteristics of these encodings. To this end, we are actively exploring theoretical approaches to better understand the connections between graph diameter, structural encodings, and model performance.
> >
> >    We are also conducting follow-up studies to investigate the intrinsic properties of RRWP, GKSE, and MKSE. The reviewer’s question has provided valuable direction and inspiration for these ongoing efforts, and we sincerely thank you for your insightful observations. These investigations aim to strengthen the theoretical underpinnings of our claims and further elucidate the unique contributions of each encoding.
> >
> > Once again, we deeply appreciate the reviewer’s continued engagement and thoughtful feedback, which have significantly enriched the clarity and rigor of our work.

---

> > > ### Comment · Reviewer_tZUc · 2024-12-03
> > >
> > > Thank you for your response. I appreciate your commitment to explaining the efficacy of GKSE and MKSE. I will maintain my score, as it is already high.

---

> > > > ### Author Response · Authors · 2024-12-03
> > > >
> > > > We sincerely thank you for your thoughtful and encouraging comments on our work. We deeply appreciate your recognition of the motivation and theoretical aspects behind our proposed GKSE and MKSE, as well as the effort we put into extending these concepts for practical application in the AI domain. Your detailed feedback on the interpretation of our experimental results has been invaluable, allowing us to enhance the quality of our manuscript significantly. Moreover, your insights have provided us with excellent directions for future research. Thank you once again for your constructive input and for maintaining a high score for our work—it truly means a great deal to us.

---

### Author Response · Authors · 2024-11-21
**Official Comment by Authors to all reviewers**

We sincerely thank the reviewers for their thorough evaluation and thoughtful feedback on our manuscript. Below, we provide a summary of our contributions, clarify key points, and address overarching themes raised in the reviews.

In this work, we proposed novel random-walk-based structural encodings (SEs) for graph transformers, named Green Kernel Structural Encoding (GKSE) and Martin Kernel Structural Encoding (MKSE). These SEs are inspired by stochastic process theory, specifically kernels used to analyze long-term behavior in random walks. While these abstract mathematical tools provide strong theoretical foundations, their direct application to real-world graph data is challenging due to the significant gap between abstract mathematical theory and the practical complexities of real-world graphs. To address this challenge, we carefully designed SEs that retain the mathematical integrity of these kernels while ensuring their adaptability to graph datasets from diverse domains. Moreover, we proposed efficient computational methods for these SEs, making them practical for large-scale applications. In addition to their practicality, we rigorously demonstrated that GKSE and MKSE possess sufficient expressiveness compared to existing SEs, further validating their utility in capturing intricate graph structures effectively.

Our method has demonstrated state-of-the-art (SOTA) performance on 7 out of 8 benchmark datasets across various graph-level tasks, showcasing its versatility. Notably, our approach excels in molecular and circuit graph domains, where it effectively captures intricate graph structural properties. To understand the reasons behind this performance, we explored unique structural characteristics of these graph types using novel observational methods, which we report as part of this work. The performance improvement is especially pronounced in large-scale datasets. For instance, on the PCQM4Mv2 dataset, our SEs exhibited a significant performance gain, highlighting their effectiveness in capturing complex structural information in real-world graph data.

We thank the reviewers for their insightful comments, which have helped us refine our manuscript and identify promising directions for future research. We have made substantial revisions to address specific concerns, clarified our contributions, and added new analyses and citations to strengthen the paper. We believe that our work makes a meaningful contribution to the field by combining rigorous mathematical theory with practical advancements in graph representation learning, and we hope our responses adequately address your concerns.

Once again, we express our gratitude for the opportunity to improve our work through your constructive feedback.

---

### Comment · Area_Chair_RD6N · 2024-11-23
**Reminder: Please Review Author Responses**

Dear Reviewers,

As the discussion period is coming to a close, please take a moment to review the authors’ responses if you haven’t done so already. Even if you decide not to update your evaluation, kindly confirm that you have reviewed the responses and that they do not change your assessment.

Thank you for your time and effort!

Best regards, AC

---

### Author Response · Authors · 2024-11-28

We sincerely thank reviewers for their thorough feedback and constructive suggestions, which have been instrumental in refining. Below, we summarize the key aspects of the reviews under major contributions, major concerns, and questions raised, along with the corresponding actions we have taken.

__Major Contributions__

1. __Novel Structural Encodings (GKSE and MKSE):__
Several reviewers appreciated our proposal of GKSE and MKSE as novel structural encodings derived from Green and Martin kernels, adapted from theoretical stochastic processes. These encodings provide a mathematically grounded and unique approach to capturing structural information in graphs.
_(Reviewers tZUc, 4iMa, ecJe)_

2. __Extensive Evaluation Across Diverse Graph Domains:__
The evaluation of GKSE and MKSE on a wide range of datasets, including molecules, circuits, and image-based graphs, was commended. Reviewers noted that our methods demonstrated SOTA performance on seven of the eight benchmark datasets.
_(Reviewers ecJe, 4iMa, jhQq)_

3. __Theoretical and Empirical Validation of Expressiveness:__
Reviewers highlighted our analysis demonstrating that GKSE and MKSE extend the expressiveness of RRWP and shortest path encodings by capturing nuanced graph structural features.
_(Reviewers tZUc, 4iMa)_


__Major Concerns and Actions Taken__

1. __Persuasiveness of Observations (Reviewer tZUc):__
_Reviewer tZUc_ expressed doubts about whether our empirical observations convincingly explain the performance improvements of GKSE and MKSE over RRWP.

   + __Action Taken:__
   We clarified in the manuscript (Line 504) that our observations are not proofs but empirical insights suggesting possible explanations for performance gains.
   Additional analysis on OCB datasets was included, showing how sparse representations in RRWP for smaller-diameter graphs may weaken performance, whereas GKSE maintains denser representations, leading to improved results.

2. __Efficiency of SEs (Reviewer 4iMa):__
Questions were raised about the memory and computational efficiency of GKSE and MKSE compared to other SEs like RRWP, shortest path encodings (SPD), and Laplacian Eigenvectors (LapEig).

   + __Action Taken:__
   We provided complexity analyses and practical runtime comparisons in Section 5.3. For example, precomputing GKSE is nearly identical to RRWP in time and space complexity, while MKSE has a slightly higher computational cost but delivers richer structural information.
   A comparison table summarizing the computational complexities of GKSE, MKSE, RRWP, SPD, and LapEig was added.

3. __Generality Beyond GRIT (Reviewers 4iMa, jhQq):__
Reviewers questioned whether GKSE and MKSE generalize beyond GRIT, as our experiments initially focused on this architecture.

   + __Action Taken:__
   We integrated GKSE and MKSE into additional architectures like GPS and CKGConv. While results with GPS were comparable, incorporating these SEs into CKGConv showed improvements across multiple datasets.
   These findings demonstrate that GKSE and MKSE are not limited to GRIT.

4. __Practical Applicability and Value of SEs (Reviewer ecJe):__
Why should practitioners prefer GKSE and MKSE over RRWP, especially given marginal improvements on certain datasets?

   + __Action Taken:__
   Experiments on subsets of the PCQM4Mv2 dataset, focusing on graphs with specific structural properties (e.g., benzene rings), showed that GKSE outperforms RRWP in capturing non-aperiodic substructures.

5. __Justification for Computational Costs (Reviewer ecJe):__
Why are GKSE and MKSE worth the slightly higher precomputation costs compared to RRWP?

   + __Action Taken:__
   We emphasized that the precomputation phase minimally impacts overall training time and that GKSE consistently delivers better results on molecular and circuit graphs. Practical runtime data was provided for large-scale datasets, showcasing that GKSE is only marginally slower than RRWP.


__Questions Raised and Responses__

1. __Why Inconsistent Baselines Across Benchmarks? (Reviewer 4iMa):__

   + __Response:__
   Missing results for GraphSAGE stemmed from unavailable hyperparameters in prior works. To address this, we conducted additional experiments for GraphSAGE on peptides and PCQM4Mv2 datasets. Results confirmed that GKSE and MKSE remain competitive.

2. __Can GKSE and MKSE Be Combined? (Reviewer 4iMa):__

   + __Response:__
   Experimental results showed no significant performance gains from combining GKSE and MKSE due to overlapping information. We recommend selecting SEs based on dataset characteristics.

3. __Connections to Hitting and Commute Times? (Reviewer ecJe):__

   + __Response:__
   While traditional Green kernels connect to hitting/commute times, our finite-step adaptations are distinct. We acknowledged this in the manuscript and highlighted it as a promising direction for future work.

---

### Meta-Review · Area_Chair_RD6N · 2024-12-08

**Metareview:**

The paper proposes two structural encodings for graph transformers, GKSE and MKSE, derived from Green and Martin kernels. These methods extend the standard form of random walks. The authors analyze some theoretical aspects of these structural encoding and evaluate their performance in combination with GRIT across multiple graph datasets. The results indicate that GKSE and MKSE have potential advantages as structural encodings.

The strengths of this paper cover the rigorous formulation and the clear presentation of the proposed methods. Empirical experiments also show some improvements. The weaknesses of this paper include the limited evaluated backbone (only GRIT), the comparison with existing SE/PE and the marginal improvement compared to the standard random-walk structural encoding RRWP. In the rebuttal, the authors provided the application of other backbones. However, the marginal improvement over the RRWP is still there.

After carefully reading the paper, I have some similar concerns about the novelty of this work, aligned with two reviewers who recommended a rejection. Let me explain this.

**Limited Novelty in Structural Encoding**
The GKSE encoding, as defined in Eq. (13), shares the same polynomial basis with RRWP for finite $K$. A linear transformation can map RRWP to GKSE, rendering the latter not fundamentally different, also limited empirical performance gain observed in the experiments. The proposed GKSE follows a formulation similar to PageRank-based random walks (with 0.5 transportation probability), which have already been explored as structural encodings in works like [1]. Consequently, the empirical performance improvements over RRWP are marginal. Even for infinite $K$, GKSE reduces to the pseudo-inverse of the graph Laplacian, effective resistance, a concept utilized as structural encoding in [2, 3]. Hence, the novelty of this contribution is marginal.

**Expressive Power Analysis** Both GKSE and MKSE are spectrum-based encodings, whose upper-limit expressive power has been extensively analyzed. Prior works [4, 5] have shown that spectral invariants are not more expressive than 3-WL, and subsequent research [6] established that they are strictly less powerful than 3-WL. The distance-generalized 1-WL, as outlined in Eq. (19), was first proposed in [1], where RRWP was also firstly introduced as an enhancement for GNNs’ expressive power. Moreover, as spectrum-based encodings, GKSE and MKSE can be seen as mappings of Laplacian PEs, aligning with stable forms proposed in [7]. The above studies [1-7] seemed to be missed in the current work discussion, leaving gaps in the contextualization of this work within the broader literature.


Given the above concerns, I lean towards rejecting this paper. While the authors provide a rigorous exposition and some empirical validation, the lack of substantial novelty, the marginal performance improvements, and the limited contextualization of prior work diminish its overall contribution. My concerns align with the two reviewers who provided weak rejections. Although Reviewer 4iMa increased the score to acceptance after the authors provided experiments on backbones other than GRIT, the performance gains remain minor, reaffirming the theoretical concerns. Moreover, Reviewer 4iMa also raised concerns regarding the computational efficiency of the proposed SEs compared to existing SEs/PEs. The authors presented a table suggesting that Lap PEs and shortest-path distances have a complexity of $O(V^3)$. However, this comparison is unfair. Under their original definitions, which involve infinite steps, GKSE and MKSE also require approximately $O(V^3)$. The reported reduction to $O(K|V||E|)$ arises solely from a $K$-step approximation. If such approximations are permissible, then $K$-step shortest-path distances would also reduce to $O(K|V||E|)$, and Laplacian PEs could achieve the same complexity by using only a subset of eigenvectors.

I still see some valid contributions from this work. However, the above concerns have to be addressed for qualified acceptance, and this may need some major revision of the manuscript. One minor thing is about the reference format which does not follow the template of ICLR. Please kindly fix this too.


[1] Li, et al., Distance encoding: Design provably more powerful neural networks for graph representation learning, NeurIPS 2020.

[2] Velingker, et al., Affinity-Aware Graph Networks, NeurIPS 2023.

[3] Zhang, et al., Nested Graph Neural Networks, NeurIPS 2021.

[4] Lim, et al., Sign and basis invariant networks for spectral graph representation learning, ICLR 2023.

[5] Fürer, On the power of combinatorial and spectral invariants, Linear Algebra and its Applications, 2010.

[6] Zhang, et al., A Complete Expressiveness Hierarchy for Subgraph GNNs via Subgraph Weisfeiler-Lehman Tests, ICML 2023.

[7] Huang, et al., On the Stability of Expressive Positional Encodings for Graphs, ICLR 2024.

**Additional Comments On Reviewer Discussion:**

My above comments have pointed out the concerns raised by the reviewers, and which concerns have been addressed and which are still there.

---

### Decision · Program_Chairs · 2025-01-22

Reject